# VectorGym: A Multi-Task Benchmark for SVG Code Generation and Manipulation

## Abstract

We introduce VectorGym, a multi-task benchmark for evaluating Vision-Language Models (VLMs) on Scalable Vector Graphics (SVG) code generation and manipulation. VectorGym addresses the critical lack of challenging benchmarks aligned with real-world design workflows, specifically requiring mastery of complex primitives and multi-step edits. Our benchmark comprises four complementary tasks: the novel Sketch2SVG (VG-Sketch) conversion; a new SVG editing dataset (VG-Edit) involving higher-order primitives and semantic reasoning; and rigorous benchmarks for Text2SVG (VG-Text) and SVG captioning (VG-Cap). VectorGym derives particular value from expert human-authored SVG annotations across all tasks, ensuring a rigorous challenge. VectorGym also introduces a VLM-as-judge metric tailored for SVG generation, validated against human judgment. Our comprehensive evaluation of leading VLMs and our own GRPO-trained models reveals significant performance gaps, establishing VectorGym as a robust framework for advancing visual code generation.

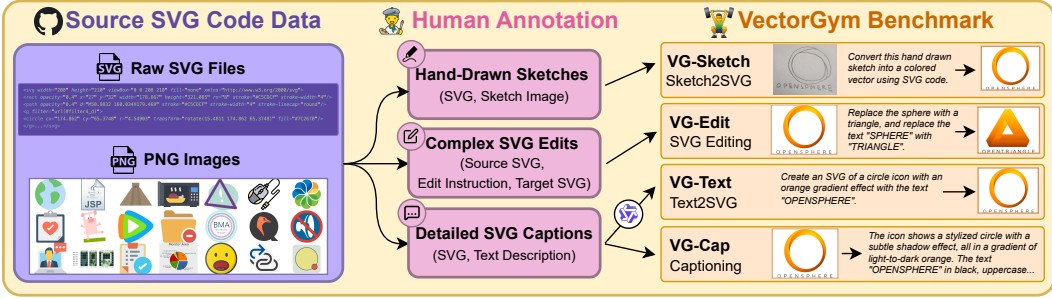

Figure 1: **Overview of VectorGym Benchmark.** VectorGym is a suite of human-annotated datasets covering Sketch2SVG (**VG-Sketch**), SVG Editing (**VG-Edit**), Text2SVG (**VG-Text**), and SVG Captioning (**VG-Cap**). Unlike prior benchmarks, it is built from diverse real-world SVGs sourced from GitHub. Human experts annotate each SVG by hand-drawing sketches, creating complex edits, and writing detailed text descriptions, which are further cleaned and adapted into instruction-style prompts at varying levels of detail. We evaluate state-of-the-art models in VectorGym.

## 1 Introduction

Scalable Vector Graphics (SVG) (Ferraiolo et al., 2000; Quint, 2003) are widely used across the web, design tooling, and digital media. Unlike raster images (Rodriguez et al., 2023b;c; Rombach et al., 2021), SVGs are programs: their code exposes geometry, style, and structure, enabling precise editing, scalable rendering, and semantic manipulation. Evaluating models on SVG therefore requires not only visual understanding but also reliable, syntax-aware code generation.

Despite rapid progress in Vision-Language Models (VLMs), existing evaluations of SVG generation remain limited. Prior datasets often target icons or basic shapes, rely on synthetic programmatic edits, rarely assess sketch-conditioned generation nor provide human gold labels (Rodriguez et al., 2023a; Wu et al., 2023; Zhang et al., 2023; Xing et al., 2025; Yang et al., 2025; Rodriguez et al.,

Table 1: **SVG Datasets Comparison.** We compare similar datasets across annotation quality, task complexity, and data source. We report the SVG type, total samples, whether each dataset supports multiple tasks, the level of SVG primitive coverage beyond simple paths (i.e. including higher order primitives, like circle, text, gradients or animation logic), and whether it includes annotations for edits and sketches, plus whether these annotations reach expert human quality with complex intent. **VectorGym** is built from real SVGs collected from GitHub (*in-the-wild*), preserving original structure and primitive detail, and *provides targets created by expert humans* with the goal of producing *complex annotations* that capture semantic understanding, design intent, and multi step reasoning.

| Dataset | SVG Type | # Samples | Multi-Task | Primitives | Edits | Sketches | Human |
|---|---|---|---|---|---|---|---|
| SVGBench (Rodriguez et al., 2023a) | *In-the-wild* | 10k | ✗ | ✓ | ✗ | ✗ | ✗ |
| VGBench (Zou et al., 2024) | Emojis | 9.5k | ✓ | ✗ | ✗ | ✗ | ✗ |
| SVGEditBench (Nishina & Matsui, 2024) | Emojis | 1.6k | ✗ | ✗ | ✓ | ✗ | ✗ |
| SVGEditBenchV2 (Nishina & Matsui, 2025) | Emojis | 1.6k | ✗ | ✗ | ✓ | ✗ | ✗ |
| UniSVG (Li et al., 2025) | Icons | 525k | ✓ | ✗ | ✗ | ✗ | ✗ |
| SVGenius (Chen et al., 2025) | Icons | 100k | ✓ | ✗ | ✓ | ✗ | ✗ |
| **VectorGym (ours)** | *In-the-wild* | 7k | ✓ | ✓ | ✓ | ✓ | ✓ |

2025). As a result, the field lacks a unified, realistic benchmark that stresses, visual understanding, vector generation and structured SVG code manipulation.

We introduce **VectorGym**, a new comprehensive multi-task benchmark for SVG generation and manipulation spanning four tasks: (1) **Sketch2SVG**(*VG-Sketch*), converting rough sketches to clean vector code; (2) **SVG Editing**(*VG-Edit*), applying natural-language edits to existing SVGs; (3) **Text2SVG** (*VG-text*), generating SVGs from text; and (4) **SVG Captioning** (*VG-Cap*), describing SVG content. VectorGym introduces Sketch2SVG and releases the first dataset of complex, human-authored SVG edits; all tasks use gold-standard human annotations.

Our benchmark covers in-the-wild diversity: icons, diagrams, emojis, fonts, logotypes, and complex illustrations, sourced from SVG-Stack (Rodriguez et al., 2023a). We pair this with careful human curation to ensure realistic task difficulty. We evaluate leading proprietary and open-source VLMs, providing a clear view of current capabilities and gaps.

Our main contributions are:

1. We introduce a comprehensive multi-task benchmark for real-world SVG code generation with gold-standard human annotations across all tasks;

2. We introduce the Sketch2SVG task and the first dataset of expert authored SVG edits with complex intent, involving rich primitives and non trivial edits.

3. We design a VLM-as-judge SVG evaluation metric tailored for sketch, text, and editing tasks, validated through human correlation studies;

4. We provide extensive evaluation and analysis of current frontier VLMs across diverse SVG generation scenarios.

## 2 RELATED WORK

**Vector Graphics Generation.** Classical vectorization methods based on shape-fitting algorithms (Li et al., 2020; Vision Cortex, 2023) struggle with complex tasks beyond image vectorization. Recent neural approaches introduce learning-based components, relying on latent variable models with differentiable rendering and attention architectures (Carlier et al., 2020; Cao et al., 2023; Lopes et al., 2019), as well as sketch abstraction (Vinker et al., 2022) and text-conditioned SVG synthesis (Jain et al., 2023). However, these methods are still not general enough to support a wide range of SVG tasks.

**VLMs for SVG Generation.** Modern VLMs (OpenAI, 2023; Comanici et al., 2025) can now produce structured code from visual inputs. StarVector (Rodriguez et al., 2023a) frames SVG creation as a visual to code generation task, jointly testing visual understanding and program synthesis. Subsequent work further supports this direction (Zhang et al., 2023; Cai et al., 2023; Yang et al., 2025).

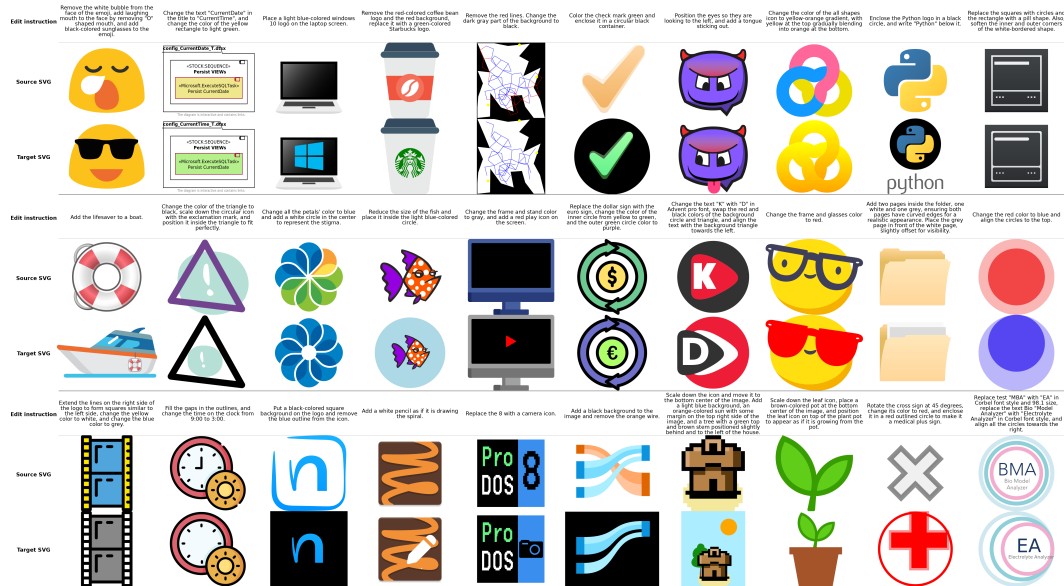

Figure 2: **Visualization of VG-Edit Test Examples.** We randomly sample 21 examples, and show the editing instruction to perform, along with the source and target vectors.

**SVG Datasets and Benchmarks.** Foundational SVG datasets include DeepSVG icons (Carlier et al., 2020), FIGR-8 (Clouâtre & Demers, 2019), and SVG-Stack (Rodriguez et al., 2023a). Several benchmarks address different SVG related tasks. UniSVG (Li et al., 2025) unifies 525k SVGs for understanding and generation. VGBench (Zou et al., 2024) aggregates multiple sources to evaluate image to SVG, text to SVG, and diagram code generation. SVGEditBench (Nishina & Matsui, 2024) and its V2 version (Nishina & Matsui, 2025) target instruction based editing using synthetic LLM generated edits or edits derived from similar SVGs. SVGenius Chen et al. (2025) covers a wide set of tasks, notably editing through algorithmic transform based operations.

Here we propose **VectorGym**, which focuses on edits created by humans following instructions that make the edits *complex* and closer to the actions of real design professionals, requiring semantic understanding. We also introduce the novel Sketch2SVG task from human drawn sketches, and we collect human validated text captions that allow evaluation of both Text2SVG and SVG captioning on realistic, high difficulty edits. See Figure 1 for a dataset comparison, and refer to Appendix A for further details.

## 3 🏋 VECTORGYM BENCHMARK

VectorGym consists of four complementary tasks that comprehensively evaluate different aspects of SVG understanding and generation. Each task is designed to assess specific capabilities while contributing to a holistic understanding of visual2code generation performance.

### 3.1 TASK DEFINITIONS

**Sketch2SVG Generation (*VG-Sketch*).** This task evaluates the ability to convert rough, hand-drawn sketches into clean SVG code. Given a bitmap sketch image with approximate shapes and imperfect lines, models must generate SVG code that captures the essential geometric structure while producing a clean, scalable vector representation. This task tests spatial reasoning, shape recognition, and the ability to abstract from noisy visual input to structured geometric primitives.

**SVG Editing (*VG-Edit*).** In this task, models are given an SVG along with an editing instruction and must produce a new SVG with the specified edit applied. VG-Edit offers unprecedented *complexity* in the challenge of SVG editing. Our editing instructions include deep understanding of the SVG

syntax, requiring the use of complex primitives like texts, animations, or color gradients. It also requires multi-step reasoning and semantic understanding (See examples in Figures 1 (right) and 2).

The challenge lies in correctly parsing the intent, identifying the relevant elements, and applying the transformation while preserving code validity, visual coherence, and the integrity of unmodified parts. Since instructions and targets were created by skilled human annotators, the edits are nontrivial, for example, adding new objects, modifying logo content or text, converting a pie chart to a bar chart, or changing facial expressions. This task evaluates both SVG structure understanding and the ability to follow complex editing instructions. Figure 2 shows examples from our test set. Unlike prior benchmarks Nishina & Matsui (2025); Chen et al. (2025), which focus on simple synthetic programmatic edits, *VG-Edit introduces* complex*, high-difficulty editing scenarios annotated by human experts.*

**Text2SVG Generation (*VG-Text*).** Given natural language descriptions of visual content, models must generate complete SVG code that accurately represents the described objects, scenes, or abstract concepts. Descriptions range from simple geometric shapes ("red circle with blue border") to complex illustrations ("minimalist icon of a house with a tree"). This task tests creative generation capabilities and the ability to translate semantic concepts into precise geometric representations.

**SVG Captioning (*VG-Cap*).** The inverse of Text2SVG generation, this task requires models to analyze existing SVG code and generate natural language descriptions that accurately capture the visual content, style, and key characteristics. High-quality captions should describe both the semantic content ("house icon") and relevant visual properties ("minimalist style," "blue and white color scheme"). This task evaluates SVG code comprehension and visual understanding.

## 3.2 DATASET CONSTRUCTION

Our datasets are built on a carefully curated SVG collection pipeline designed to ensure diversity across content types, complexity levels, and visual styles. We source high quality and diverse SVGs from the SVG Stack dataset (Rodriguez et al., 2023a), an established collection that includes icons, diagrams, emojis, fonts, logotypes, and complex illustrations. Since the original data was extracted from GitHub, it naturally reflects in the wild SVG code, including higher order primitives such as text, gradients, polygons, and animations. This makes the dataset more representative of real design workflows and provides challenging examples for model development.

Our automatic curation builds on insights from prior SVG datasets (Carlier et al., 2020; Clouâtre & Demers, 2019; Nishina & Matsui, 2024; Li et al., 2025; Chen et al., 2025). We extracted 7,000 candidate samples from the SVG Stack training split through multi stage filtering, including token length constraints (2k to 8k tokens to retain meaningful complexity), color entropy thresholding (normalized entropy greater than 0.55), and random subsampling followed by human visual inspection. After filtering, the final training set contains 6.5k samples. From these, we selected 100 samples to form our validation set, used for method tuning, in context learning, human evaluation, and metric design (see Section 3.3). We applied the same pipeline to produce the test split to obtain 300 samples, sourced from the SVG-Stack test set.

**Human Annotation Process.** We partnered with two specialized data annotation vendors to produce high quality annotations across sketch and editing tasks. The process involved more than 20 annotators with diverse backgrounds and expertise in design, vector graphics, and coding. Annotators were provided with drawing tools, coding utilities, and curated SVG collections to perform edits and create sketches on different surfaces. They were specifically instructed to produce challenging edits, involving multi-step reasoning, and real design intent, and we iterated several times on these samples to validate their complexity and quality. See Appendix A.1 for full details on the annotation methodology, quality assurance procedures, and complexity requirements.

**Complex Annotations.** In our setup, *complex annotations* refer to human created editing instructions and corresponding SVG modifications that require things like deeper understanding of the SVG syntax because they introduce hiuigher order SVG primitives like texts, gradients or animations, also edits involving semantic understanding, multi step reasoning (change many things at the same time), and design intent beyond what can be achieved through simple geometric or algorithmic transformations. These annotations involve operations such as adding new objects, integrating external SVG elements, inserting text with meaningful placement, restructuring layouts, or applying

Table 2: **VLM as a Judge and Human Correlation Analysis.** We run generation on the tasks for Claude 4.5, Gemini 3 Pro, and GPT-4o, and evaluate outputs using a range of VLMs (both closed and open, large models) to score them with the prompts presented. We also collect human ratings using the same instructions given to VLM judges, then compute Pearson correlation to identify the best VLMs as judges. The evaluation uses 100 validation samples extracted from the training set. Results show Gemini 3 Pro is generally the best judge, except for the editing task where Qwen3.VL appears to be a better choice. Sketch and text tasks show lower correlations, likely due to the more creative nature of these tasks.

| Task | Generator | Models used as Judges | | | | | | |
|---|---|---|---|---|---|---|---|---|
| | | Claude 4.5 Sonnet | Gemini 2.5 Flash | Gemini 3 Pro | GPT 5.1 | Qwen2.5VL 72B | Qwen3.VL 235B | GLM4.5 355B |
| VG-Sketch | Ground Truth | 1.00 | 1.00 | 1.00 | 1.00 | 1.00 | 1.00 | -0.07 |
| | Claude 4.5 Sonnet | 0.63 | 0.73 | 0.72 | 0.62 | 0.57 | 0.69 | 0.67 |
| | Gemini 3 Pro | 0.79 | 0.82 | 0.80 | 0.78 | 0.76 | 0.79 | 0.72 |
| | GPT 4o | 0.66 | 0.70 | 0.74 | 0.61 | 0.59 | 0.72 | 0.64 |
| | *Average* | **0.77** | **0.81** | **0.81** | **0.75** | **0.73** | **0.80** | **0.49** |
| VG-Cap | Ground Truth | 1.00 | 1.00 | 1.00 | 1.00 | 1.00 | 1.00 | 1.00 |
| | Claude 4.5 Sonnet | 0.62 | 0.57 | 0.71 | 0.62 | 0.65 | 0.71 | 0.60 |
| | Gemini 3 Pro | 0.48 | 0.47 | 0.55 | 0.49 | 0.43 | 0.53 | 0.48 |
| | GPT 4o | 0.52 | 0.46 | 0.55 | 0.47 | 0.53 | 0.54 | 0.55 |
| | *Average* | **0.66** | **0.63** | **0.70** | **0.65** | **0.65** | **0.69** | **0.66** |
| VG-Edit | Ground Truth | -0.10 | 0.10 | 1.00 | 1.00 | 0.27 | 1.00 | 0.08 |
| | Claude 4.5 Sonnet | 0.29 | 0.30 | 0.49 | 0.53 | 0.28 | 0.45 | 0.48 |
| | Gemini 3 Pro | 0.49 | 0.47 | 0.54 | 0.57 | 0.04 | 0.61 | 0.56 |
| | GPT 4o | 0.59 | 0.61 | 0.61 | 0.69 | 0.29 | 0.64 | 0.62 |
| | *Average* | **0.32** | **0.37** | **0.66** | **0.70** | **0.22** | **0.67** | **0.43** |
| VG-Text | Ground Truth | 0.01 | -0.07 | -0.08 | 0.19 | -0.19 | 0.15 | -0.07 |
| | Claude 4.5 Sonnet | 0.16 | 0.43 | 0.58 | 0.21 | 0.15 | 0.23 | 0.08 |
| | Gemini 3 Pro | 0.37 | 0.42 | 0.44 | 0.48 | 0.24 | 0.37 | 0.32 |
| | GPT 4o | 0.50 | 0.71 | 0.63 | 0.58 | 0.25 | 0.66 | 0.55 |
| | *Average* | **0.26** | **0.38** | **0.40** | **0.37** | **0.11** | **0.35** | **0.22** |

several coordinated edits simultaneously. They reflect realistic design actions performed by human experts and cannot be reproduced by rule based procedures or low level manipulations.

### 3.3 DESIGNING A VLM-AS-JUDGE EVALUATION METRIC FOR SVG GENERATION

Traditional evaluation metrics for SVG generation (typically based on image reconstruction or text–image alignment) often fall short in capturing the nuanced visual and semantic qualities that determine the success of generated vector graphics (Rodriguez et al., 2023a; Li et al., 2025; Chen et al., 2025). Existing work lacks comprehensive evaluation frameworks tailored to SVG generation, particularly metrics that can jointly assess visual fidelity and semantic alignment in vector code outputs (Zou et al., 2024; Nishina & Matsui, 2025).

*VLM-as-judge* (VLMAJ) metrics have become popular because they provide strong supervision signals for subjective task assessments, especially in text and image generation tasks Mañas et al. (2024). *Existing VLMAJ metrics do not capture the nuances of SVG code and SVG rendering.* They are also not reliable for tasks such as sketch based generation and SVG editing, where no consistent metric previously existed. For this reason we design a metric specifically tailored to the four SVG generation tasks in our benchmark.

We generate outputs from several strong baseline models and then apply carefully designed prompts to a set of powerful VLMs, both open and closed source, to obtain scores from 0 to 5 following clear evaluation criteria (see Appendix D). We run the same evaluation setup with human raters and then compute Pearson correlations between VLM and human scores. This produces four task specific VLMAJ metrics, one for each task in our benchmark, providing a more faithful measure of instruction following, SVG structural correctness, and semantic alignment.

**1. Metric Development Process.** We carefully develop task-specific evaluation prompts designed to guide VLMs in assessing different aspects of SVG generation quality. For each of the four main generation tasks, we craft specialized prompts that encourage models to evaluate: (1) visual accuracy and fidelity; (2) semantic alignment with input requirements; (3) code quality and efficiency; and (4) overall aesthetic appeal.

**2. Judge Model Selection.** To identify the most reliable VLM judge, we conduct a systematic comparison across state-of-the-art models: Claude 4.5 Sonnet, Gemini 2.5 Flash, Gemini 3 Pro,

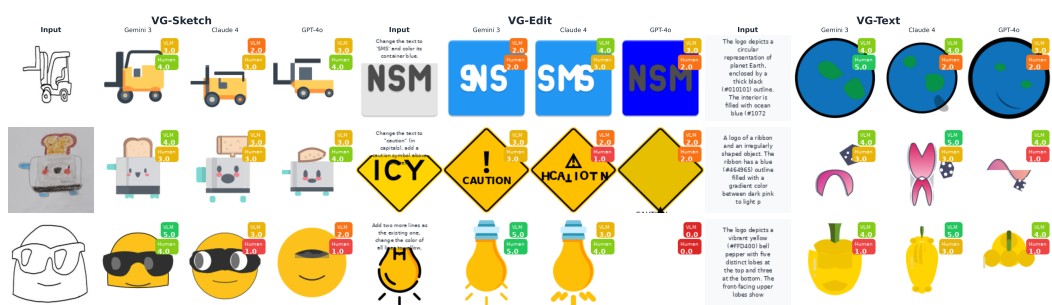

Figure 3: **Qualitative results on VectorGym.** We display VLM-Judge and Human scores on a scale from 0 to 5. Each task shows three validation samples alongside the strongest models in our evaluation. Human ratings tend to be stricter, while VLM judges are more permissive and often cluster around mid-range values when uncertain.

Qwen 2.5VL 72B-235B, and GLM 4.5 355B, covering closed-open source performance, and large-mid scale sizes.

**3. SVG Generation and VLMAJ Evaluation.** We evaluate state of the art models on the validation set (100 samples). We select Claude 4.5 Sonet, Gemini 3 Pro, and GPT4o, and run generation experiments on the four tasks. The resulting outputs are then scored by all VLM judges described above. We also compute scores for the ground truth SVGs, which should receive the highest ratings, providing a way to assess the overall dataset quality.

**4. Human Evaluation.** We repeat the same evaluation setup with human raters. They receive the same prompt with the specified criteria and score the generations from all models as well as the ground truth data. A total of 17 human evaluators participated, all technical engineers or AI and design experts, producing around 674 ratings used to correlate each VLM with human judgment.

**5. Correlation Validation and VLMAJ Selection.** We compute Pearson correlation coefficients between human judgments and each candidate VLM judge for every task and report the results in Table 2. We also include average validation scores for the three generation models in Table 6, showing both human ratings and VLM evaluations. Ground Truth acts as a reliable anchor only for VG Sketch and VG Cap, where human agreement is high due to clearer visual semantics. For VG Edit and VG Text, correlations drop even on perfect examples, indicating that these tasks contain more structural ambiguity and are inherently harder to evaluate with full consensus. This further motivates the need for robust automatic judges tailored to each task. The correlation results highlight clear preferences among VLM judges. Gemini Flash and Gemini 3 Pro provide the strongest alignment with human ratings in VG Sketch, and Gemini 3 Pro also achieves the highest correlation in VG Cap. For VG Edit, which is the most challenging task, Gemini 3 Pro and GPT 5.1 stand out as the only reliable options, with GPT 5.1 showing a slight advantage. For VG Text, Gemini Flash ranks highest, with GPT 5.1 again performing consistently. Qwen3 VL 235B emerges as the most stable open source option, performing well across VG Sketch, VG Cap, and VG Edit, with the main weakness appearing in VG Text. Based on these findings, we select Gemini 3 Pro as the primary VLMAJ judge for VG Sketch, VG Cap, and VG Text. For VG Edit, we use GPT 5.1, which shows the strongest alignment with human judgments on this task.

### 3.4 EVALUATION

We describe the metrics used for evaluation in VectorGym, in addition to the VLM-as-Judge metric defined above.

**Visual Similarity.** For tasks that require visual reproduction (Sketch2SVG, Text2SVG), we measure similarity between generated and target SVGs after rendering them to pixels. We use pixel Mean Squared Error (MSE), perceptual similarity (LPIPS), and Dino, a deep feature metric that captures alignment in learned representations (Oquab et al., 2023).

**Semantic Accuracy.** For Text2SVG, we evaluate whether the generated SVG captures the intended semantic meaning of the text through CLIP-based similarity and the VLM-Judge metric. For SVG

Editing, we rely exclusively on the VLM-Judge since CLIP does not align well with editing instructions or edited outputs.

**SVG Captioning Metrics.** For captioning, we report ROUGE-L F1 (0 to 100, higher is better), BGE-M3 cosine similarity (0 to 100, higher is better), and an LLM-based rubric score (GPT-5 mapped from 0 to 5 into 0 to 100). Metrics are computed pairwise over each reference and prediction caption, then averaged across the corpus.

**Human Evaluation.** A subset of outputs from the top performing models on the validation split is evaluated by expert annotators. They assess overall quality, semantic correctness, and task specific criteria (see Table 6).

**Overall VectorGym Score.** We define an overall score for our benchmark, intended to measure multi-task performance across SVG generation from sketches and texts, complex editing of SVGs, and SVG understanding through captioning from code. First, we compute a task-specific score $S_{task}$ for each of the four tasks. For Sketch2SVG and SVG Editing, the score is the average of the VLM Judge, DINO, inverted MSE $(100 - MSE)$, and inverted LPIPS $(100 - LPIPS)$, ensuring all components contribute positively. For Text2SVG, we average the VLM Judge, CLIP, and DINO scores. For SVG Captioning, we average the VLM Judge, BGE-M3, and ROUGE scores. Finally, the overall VectorGym score is computed as the arithmetic mean of the four task-specific scores:

$$\text{VectorGym} = \frac{1}{4} \sum_{\tau \in \mathcal{T}} S_\tau \tag{1}$$

where $\mathcal{T} = \{\text{Sketch}, \text{Edit}, \text{Text}, \text{Caption}\}$. All individual metrics are scaled to a range of $[0, 100]$ prior to aggregation.

## 4 EXPERIMENTS

We conduct comprehensive evaluation across all four VectorGym tasks using state-of-the-art VLMs. Our experimental setup is designed to provide fair comparison while highlighting the unique challenges of SVG code generation.

### 4.1 METHODS AND BASELINES

We conduct a comprehensive evaluation using all available state-of-the-art VLMs that support code generation capabilities. Our baseline selection follows a systematic approach to ensure comprehensive coverage of the current landscape.

**In-Context Learning Experiments.** First we evaluate the capabilities of frontier trained models capanilities at this tasks with in-context learning giving a strong prompt to describne the task to perform. We include open and closed source models wioht the prompts specifgied in Appendix D.

*A. Closed-Source Models.* We evaluate leading commercial VLMs that demonstrate strong performance on visual understanding and code generation tasks: Gemini 2.5 Flash, Gemini 3 Pro, GPT4o, GPT-5.1, and Claude Sonet 4.5. These models represent the current state-of-the-art in multimodal understanding and have shown exceptional capabilities in various vision-language and code generation benchmarks.

*B. Open-Source Models.* To ensure comprehensive coverage and reproducible research, we include leading open-source alternatives: Qwen2.5VL 32B-72B Instruct, Qwen3VL 8B-235B, and GLM4.5V 108B. We made best efforts to identify and include all available VLM models with public code implementations that could be executed on our tasks.

**RL Training Experiments.** We also train a Qwen3VL 8B Instruct model using the RLRF (Reinforcement Learning from Rendering Feedback) framework (Rodriguez et al., 2025), which applies GRPO (Shao et al., 2024) together with rendered SVG outputs to compute rewards. The model is trained on the VectorGym train split across all four tasks simultaneously. Further details on this approach are provided in Appendix C.

(a) **VG-Sketch Qualitative Results.** The leftmost column displays the input raster sketch, followed by the outputs from top-performing models. Gemini 3 Pro demonstrates superior fidelity in preserving topological structure compared to GPT-5.1 and others.

(b) **VG-Edit Qualitative Results.** Left to right: natural language edit instruction, input SVG, and model outputs. Gemini 3 Pro, Claude 4.5 Sonnet, and GPT5.1 effectively execute complex semantic modifications, whereas our trained models struggle to follow some multi-step edits.

Figure 4: Qualitative comparison of model performance on Sketch2SVG and SVG Editing tasks.

## 5 RESULTS

We present a comprehensive evaluation of state-of-the-art VLMs across the four VectorGym tasks. Our analysis reveals significant performance variance across different modalities of SVG generation and manipulation, highlighting distinct capability gaps between proprietary and open-source models.

Table 3: **Sketch2SVG and SVG Editing Performance.** Metrics are reported such that higher values indicate better performance (↑) or lower values indicate better performance (↓). To compute the unified **Score**, MSE and LPIPS are inverted ($100 - x$) and averaged with VLM Judge and DINO, all scaled to $[0, 100]$. **Overall** represents the arithmetic mean of scores across all four tasks. The best results in each category are marked in **bold**.

| Model | Sketch2SVG | | | | | SVG Editing | | | | | Overall |
|---|---|---|---|---|---|---|---|---|---|---|---|
| | VLM J ↑ | MSE ↓ | DINO ↑ | LPIPS ↓ | Score ↑ | VLM J ↑ | MSE ↓ | DINO ↑ | LPIPS ↓ | Score ↑ | VectorGym ↑ |
| *Open-source Models* | | | | | | | | | | | |
| Qwen2.5VL 72B Instruct | 12.80 | 16.43 | 69.87 | 43.95 | 55.57 | 16.60 | 18.68 | 70.35 | 38.21 | 57.52 | 44.27 |
| Qwen2.5VL 32B Instruct | 17.80 | 15.15 | 71.63 | 42.65 | 57.91 | 20.20 | 17.04 | 72.31 | 37.05 | 59.61 | 49.16 |
| GLM4.5V | 33.80 | 14.14 | 78.61 | 41.35 | 64.23 | 37.60 | 13.39 | 80.90 | 31.76 | 68.34 | 57.02 |
| Qwen3VL 8B Instruct | 33.00 | 13.76 | 81.01 | 40.97 | 64.82 | 57.40 | 11.01 | 90.44 | 25.27 | 77.89 | 58.74 |
| Qwen3VL 235B Instruct | 40.00 | 13.37 | 83.69 | 40.23 | 67.52 | 60.40 | 9.02 | 91.17 | 22.11 | 80.11 | 62.32 |
| Qwen3VL 8B Gym (Ours) | **46.00** | **11.99** | **88.25** | **39.37** | **70.72** | **67.00** | **8.36** | **93.94** | **21.34** | **82.81** | **66.05** |
| *Proprietary Models* | | | | | | | | | | | |
| Gemini 2.5 Flash | 36.80 | 13.67 | 79.13 | 40.45 | 65.45 | 65.80 | 9.98 | 90.54 | 21.16 | 81.30 | 61.42 |
| GPT-4o | 46.00 | 13.17 | 85.11 | 39.74 | 69.55 | 66.80 | 8.43 | 92.27 | 21.24 | 82.35 | 64.93 |
| Claude Sonnet 4.5 | 58.80 | 12.54 | 88.42 | 39.29 | 73.85 | 79.40 | 6.29 | **95.61** | 16.46 | 88.07 | 70.31 |
| GPT-5.1 | 64.00 | 12.28 | 89.47 | 38.42 | 75.69 | 78.00 | 5.92 | 95.59 | 16.83 | 87.71 | 71.36 |
| Gemini 3 Pro | **72.20** | **11.31** | **89.78** | **36.43** | **78.56** | **81.20** | **5.89** | 95.55 | **16.01** | **88.71** | **73.17** |

### 5.1 SKETCH2SVG GENERATION

The Sketch2SVG task evaluates the model's ability to infer vector geometry from raster sketches, a problem characterized by high ambiguity and visual abstraction. As shown in Table 3, **Gemini 3 Pro achieves the highest performance**, obtaining a Score of 78.56 and a VLM Judge score of 72.20. This indicates a superior capability in mapping pixel-level visual features to precise SVG path commands. GPT-5.1 follows with a Score of 75.69.

Table 4: **Text2SVG and SVG Captioning Performance.** Higher values indicate better performance (↑). DINO scores for Text2SVG are scaled to $[0, 100]$. The **Score** column represents the unweighted average of metrics within each task.

| Model | Text2SVG | | | | SVG Captioning | | | |
|---|---|---|---|---|---|---|---|---|
| | VLM J ↑ | CLIP ↑ | DINO ↑ | Score ↑ | VLM J ↑ | BGE-M3 ↑ | ROUGE ↑ | Score ↑ |
| *Open-source Models* | | | | | | | | |
| Qwen2.5-VL-72B-Instruct | 25.80 | 25.78 | 71.00 | 40.86 | 9.60 | 52.08 | 7.70 | 23.13 |
| Qwen3-VL-8B-Instruct | 55.20 | 29.48 | 81.71 | 55.46 | 25.20 | 66.27 | 18.87 | 36.78 |
| GLM-4.5V | 59.40 | 28.91 | 80.44 | 56.25 | 38.00 | 62.85 | 16.86 | 39.24 |
| Qwen3-VL-32B-Instruct | 22.60 | 24.95 | 68.96 | 38.84 | 38.40 | 66.10 | 16.35 | 40.28 |
| Qwen3-VL-235B-Instruct | 66.80 | 29.60 | 82.63 | 59.68 | **40.40** | 67.14 | 18.33 | 41.96 |
| Qwen3-VL-8B-Gym (Ours) | **72.80** | **30.55** | **87.46** | **63.60** | 35.80 | **79.76** | **25.58** | **47.05** |
| *Proprietary Models* | | | | | | | | |
| GPT-4o | 74.60 | 30.43 | 84.23 | 63.09 | 46.00 | 66.82 | 21.33 | 44.72 |
| Gemini 2.5 Flash | 54.00 | 27.67 | 77.65 | 53.11 | 45.80 | 69.24 | 22.45 | 45.83 |
| Claude Sonnet 4.5 | 89.00 | **30.91** | 87.66 | 69.19 | 59.20 | 70.17 | 21.08 | 50.15 |
| GPT-5.1 | **93.00** | 30.83 | 88.20 | **70.68** | 62.20 | 70.45 | 21.49 | 51.38 |
| Gemini 3 Pro | 89.80 | 30.87 | **89.09** | 69.92 | **70.40** | **72.27** | **23.83** | **55.50** |

Notably, the performance gap between the top model and the open-source baseline is significant. However, **our proposed Qwen3VL 8B Gym model achieves a Score of 70.72, surpassing both GPT-4o (69.55) and the much larger Qwen3VL 235B (67.52).** The Gym model's VLM Judge score (46.00) represents a substantial improvement over the base Qwen3VL 8B Instruct (33.00), validating the efficacy of curriculum learning for structural visual alignment.

## 5.2 SVG EDITING

SVG Editing requires disjoint reasoning capabilities: parsing the existing code structure and manipulating it according to natural language instructions. **Gemini 3 Pro again leads this task with a Score of 88.71**, closely followed by Claude Sonnet 4.5 (88.07). Claude Sonnet 4.5 notably achieves the highest DINO score (95.61) and lowest MSE (6.29), suggesting it generates visually faithful edits even if the structural implementation differs slightly from the ground truth.

**Our Qwen3VL 8B Gym demonstrates remarkable competitiveness in this domain**, achieving a Score of 82.81. This performance exceeds that of GPT-4o (82.35) and approaches the proprietary frontier. The low MSE (8.36) of the Gym model compared to the base 8B model (11.01) indicates that fine-tuning on edit trajectories significantly enhances the model's precision in coordinate manipulation.

## 5.3 TEXT2SVG GENERATION

Table 4 presents our Text2SVG generation results, revealing clear performance hierarchies and interesting patterns. Among proprietary models, **GPT-5.1 achieves state-of-the-art performance** with a VLM Judge score of 93.00 and an overall Score of 70.68. The proprietary models exhibit a relatively narrow performance band, with Gemini 3 Pro (69.92) and Claude Sonnet 4.5 (69.19) performing comparably.

Among open-source models, **our fine-tuned Qwen3VL 8B Gym outperforms the larger Qwen3VL 235B baseline** (Score: 63.60 vs. 59.68) and achieves parity with GPT-4o (63.09). This result emphasizes that for well-defined generation tasks, specialized smaller models can effectively compete with general-purpose frontier models.

## 5.4 SVG CAPTIONING

The SVG Captioning results in Table 4 reveal interesting patterns distinct from the generation tasks.

**Gemini 3 Pro dominates the VLM Judge metric (70.40)**, significantly outperforming other models, which aligns with its robust ability to map code structure back to high-level semantic descriptions. However, the traditional NLP metrics show different rankings: our Qwen3VL 8B Gym achieves the highest BGE-M3 (79.76) and ROUGE scores (25.58) across the entire benchmark.

**Qwen3VL 8B Gym outperforms all proprietary models in keyword-based metrics.** This discrepancy between its state-of-the-art retrieval scores and its lower VLM Judge score (35.80 compared to 40.40 for the Qwen3VL 235B baseline) suggests that while the Gym model captures salient semantic details, it may lack the conversational fluency or formatting preference favored by the VLM Judge.

## 5.5 Cross-Task Analysis

Our comprehensive evaluation across Text2SVG, SVG Editing, and Sketch2SVG reveals several critical insights about current VLM capabilities in vector graphics generation.

**Overall Performance Hierarchy.** Aggregating across all tasks, Gemini 3 Pro achieves the highest VectorGym score of 73.17, followed by GPT-5.1 (71.36). This establishes Gemini 3 Pro as the most capable model for multimodal code-visual reasoning tasks.

**Effectiveness of Specialized Fine-Tuning.** The Qwen3VL 8B Gym model achieves an overall score of 66.05, surpassing GPT-4o (64.93) and substantially outperforming its larger counterpart, Qwen3VL 235B (62.32). This finding validates the hypothesis that the limitations of smaller parameter counts can be effectively offset by high-quality, task-specific curriculum learning in the SVG domain.

**Task Complexity.** The results establish a clear difficulty hierarchy: Text2SVG (easiest, GPT-5.1: 93.00) > SVG Editing (intermediate, Gemini 3 Pro: 81.20) > Sketch2SVG (Gemini 3 Pro: 72.20) > SVG Captioning (hardest, Gemini 3 Pro: 70.40). This ranking aligns with intuitive expectations: text descriptions provide explicit semantic guidance, editing requires understanding existing structures, sketches demand interpretation of imprecise visual input, while captioning requires the rigorous abstraction of high-level semantics from low-level geometric code.

## 6 Conclusion

We introduced VectorGym, a new comprehensive multi-task benchmark for SVG code generation that encompasses Sketch2SVG, SVG editing, Text2SVG, and SVG captioning. VectorGym introduces Sketch2SVG and releases the first dataset of complex, human-authored SVG edits, with gold-standard human annotations across all tasks. Our 7,000-sample evaluation and novel VLM-as-judge metrics reveal significant performance gaps between proprietary and open-source models, with open-source alternatives showing competitive results in editing and captioning. VectorGym establishes a new evaluation standard for visual code generation and provides robust benchmarks to advance SVG generation capabilities.

**Use of LLMs** We leveraged large language models (LLMs) to support different aspects of this work. They assisted with coding tasks needed to build the datasets and run experiments. Models such as GPT-4o, GPT-5, and Claude-4-Sonnet were also used to help with related work exploration and to ensure a comprehensive literature review. In addition, we employed LLMs for rephrasing and refinement while writing this paper, with the goal of improving flow, clarity, and correcting spelling errors. Importantly, we followed strict rules to preserve the accuracy and details of our contributions, and all generated content was carefully reviewed, manipulated, and edited by the authors.

**Limitations** VectorGym expands the range of capabilities that can be evaluated and optimized for fine grained control of state of the art SVG models. We tested several leading models in a zero shot setting, and we also ran RL training experiments that produced strong results. Still, we do not fully explore the space of training strategies for these tasks. Future research can focus on improving how models tackle sketch based generation and complex editing, potentially with more efficient and more accurate approaches tailored to these settings.

**Ethics Statement**   The models evaluated in this benchmark may exhibit biases inherited from their training data, potentially affecting the fairness and representation of generated SVG content across different demographics, cultures, and artistic styles. We have performed extensive filtering and human curation to ensure VectorGym does not include such instances.

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

## A  VECTORGYM DATA CREATION

Here we provide additional details on the VectorGym datasets. Figures 2 and 5 illustrate test samples for the Sketch2SVG (VG-Sketch) and SVG Editing (VG-Edit) tasks. We further describe the annotation methodology, data creation and sampling process, annotation details, and task definitions.

### A.1  ANNOTATION METHODOLOGY

#### A.1.1  DATA CURATION AND SAMPLING

We extracted 7,000 high-quality samples from the SVG-Stack dataset through a rigorous multi-stage filtering process:

**Visual Quality Assessment:** Human experts manually reviewed SVG samples to identify visually appealing and well-formed graphics, filtering out corrupted, overly simplistic, or poorly designed samples.

**Token Length Filtering:** We applied token length constraints (2,000-8,000 tokens) to ensure meaningful complexity while maintaining computational feasibility. This range captures rich, detailed SVGs without exceeding practical processing limits for current VLMs.

**Color Entropy Thresholding:** We computed color entropy for each SVG to ensure visual diversity, filtering samples with insufficient color variation or monotonic palettes.

**Random Sampling:** Final samples were randomly selected to avoid systematic biases in content distribution.

From the curated set of 7,000 samples, we kept the 300 items that originally belonged to the SVG Stack test split as our test set to avoid any train and test contamination. We also selected 100 samples from the training split for validation, which we used during development for method tuning, and for the human evaluation and correlation study used to design our VLM as a judge metric (see Section 3.3).

#### A.1.2  ANNOTATION VENDOR PARTNERSHIP

We partnered with two specialized data annotation vendors to ensure task-specific expertise:

**Vendor 1 - Sketch and Caption Generation:** Specialized in visual content creation, responsible for sketch generation and text descriptions. Annotators were equipped with professional drawing tools (digital tablets, cameras for hand-drawn sketches) and trained on SVG visual analysis.

**Vendor 2 - SVG Editing:** Focused on technical SVG manipulation, staffed with annotators having design and vector graphics backgrounds. We developed custom SVG editing tools specifically for this project to enable precise modifications.

#### A.1.3  ANNOTATOR DEMOGRAPHICS AND TRAINING

Our annotation team comprised over 20 annotators with diverse demographics and gender representation. All annotators underwent specialized training:

**Technical Requirements:** Background in design, vector graphics, or coding. Annotators were tested on SVG understanding and tool proficiency before assignment.

**Equipment and Tools:** Professional cameras for photographing hand-drawn sketches, digital drawing tablets, custom SVG editing software, and standardized annotation interfaces.

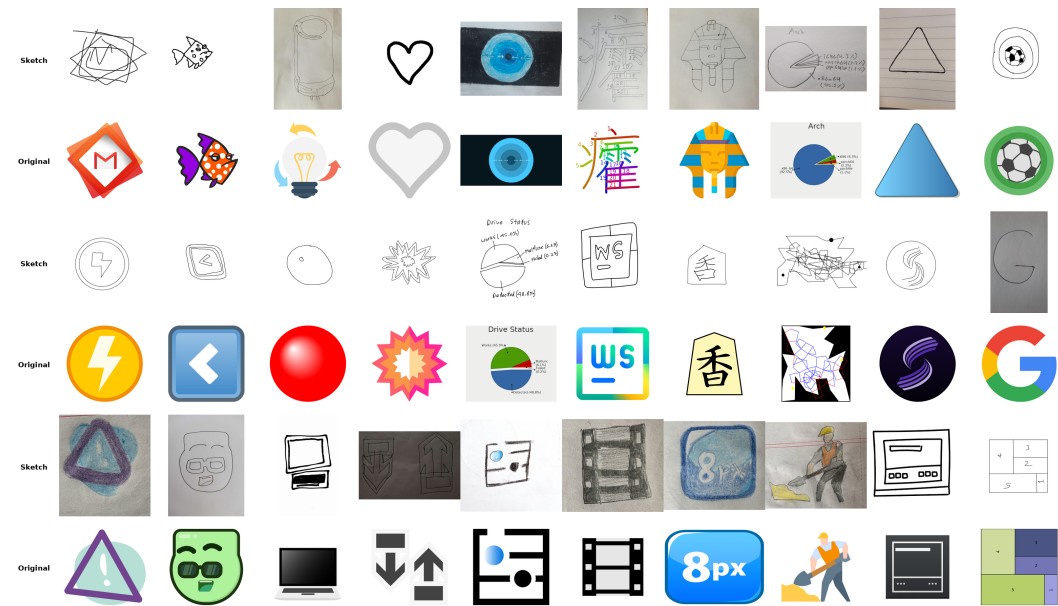

Figure 5: **Visualization of VG-Sketch Test Examples.** We randomly sample 30 examples, and show the sketch and the target vector.

### A.1.4 TASK-SPECIFIC ANNOTATION PROCEDURES

**Sketch2SVG Generation:** Annotators were provided with SVG images and asked to create corresponding sketches in two variants:

- **Hand-drawn:** Using pen or pencil on paper, photographed with standardized lighting and resolution
- **Digital:** Created using drawing tablets and stylus input for consistent digital sketches

Both variants included colored and black-and-white versions to test model robustness across different input modalities.

**SVG Editing - Ensuring Complexity:** We implemented strict complexity requirements to avoid trivial edits that could be synthetically generated:

*Prohibited Simple Edits:* Rotation, color changes, scaling, basic shape removal - operations easily automated by current LLMs.

*Required Complex Edits:* Path modifications, primitive additions, parameter adjustments, conceptual additions requiring semantic understanding. For example:

- Adding elements from other SVGs in the database (e.g., incorporating a needle shape into a hammer SVG)
- Modifying facial expressions in character illustrations
- Converting chart types (pie to bar charts)
- Structural modifications requiring new geometric primitives

**Caption Generation:** We implemented a comprehensive multi-stage process for generating high-quality text descriptions:

1. **Detailed Visual Description:** Annotators created comprehensive descriptions of vector graphics, with particular emphasis on color specification. To ensure color accuracy, annotators were required to include hexadecimal color codes in parentheses alongside natural language color descriptions (e.g., "red (#FF0000)").

2. **Cross-validation with VLM:** All human-generated descriptions were processed and cross-validated using Qwen2-VL-32B to ensure consistency and completeness of visual descriptions.

3. **Instruction Reformatting:** Captions were systematically reformatted from descriptive statements into instruction-style prompts suitable for the Text2SVG generation task. This process generated two distinct variants:

   - **Hexadecimal Color Version:** Instructions containing precise hexadecimal color specifications, which empirically demonstrate superior SVG generation accuracy

   - **Natural Language Color Version:** Instructions using standard color names for broader accessibility

4. **Quality Validation:** Final consistency checks and inter-annotator agreement measurement across all caption variants

**Quality Assurance:** All annotations underwent rigorous quality control including automated SVG syntax validation, human verification of task requirements, and consistency checks across related task pairs.

## B  ADDITIONAL QUALITATIVE RESULTS

We provide additional figures (Figures 6–10) showing qualitative results of the models on the presented tasks.

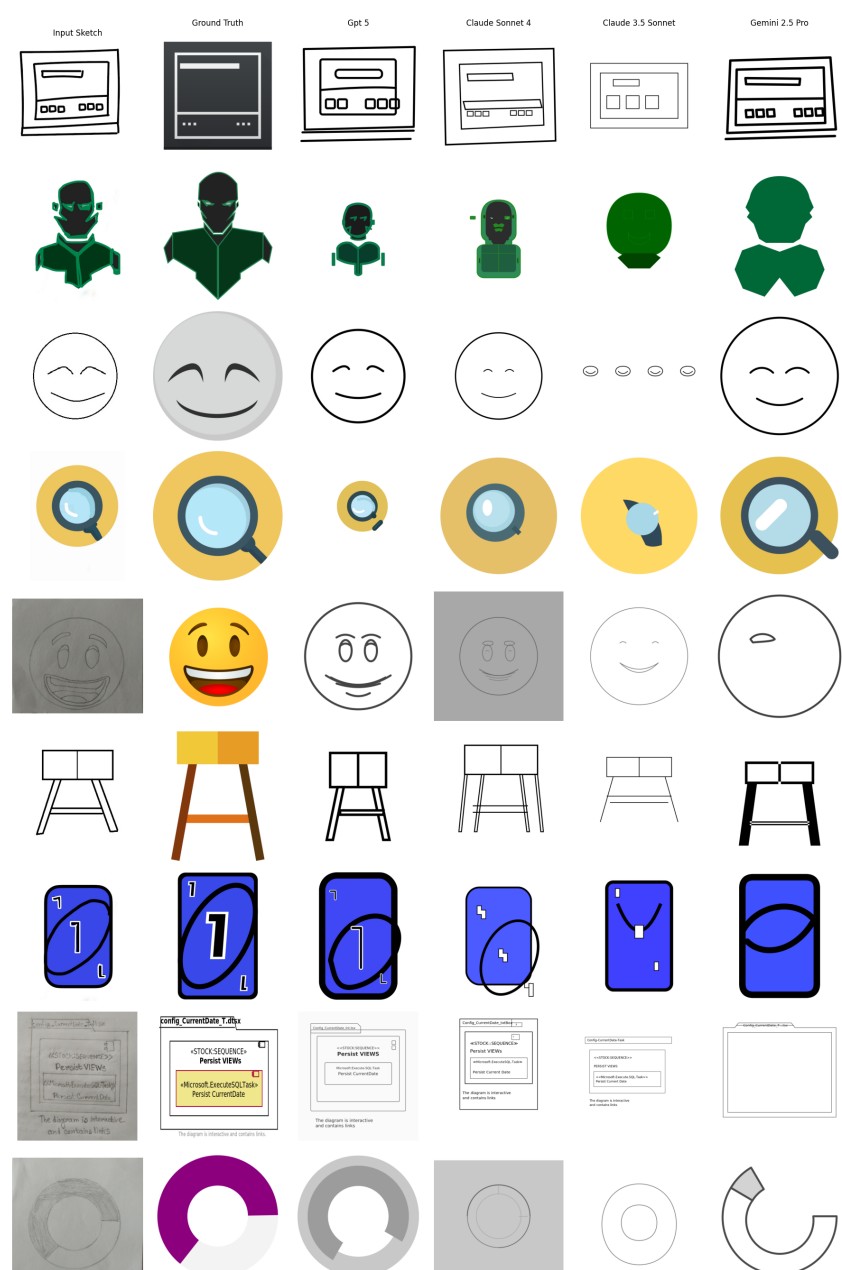

Figure 6: Visualization of test performance on the Sketch2SVG task. When the input sketch lacks color, models tend not to introduce new colors. In contrast, when color is present in the sketch, models successfully reproduce it in the generated SVG.

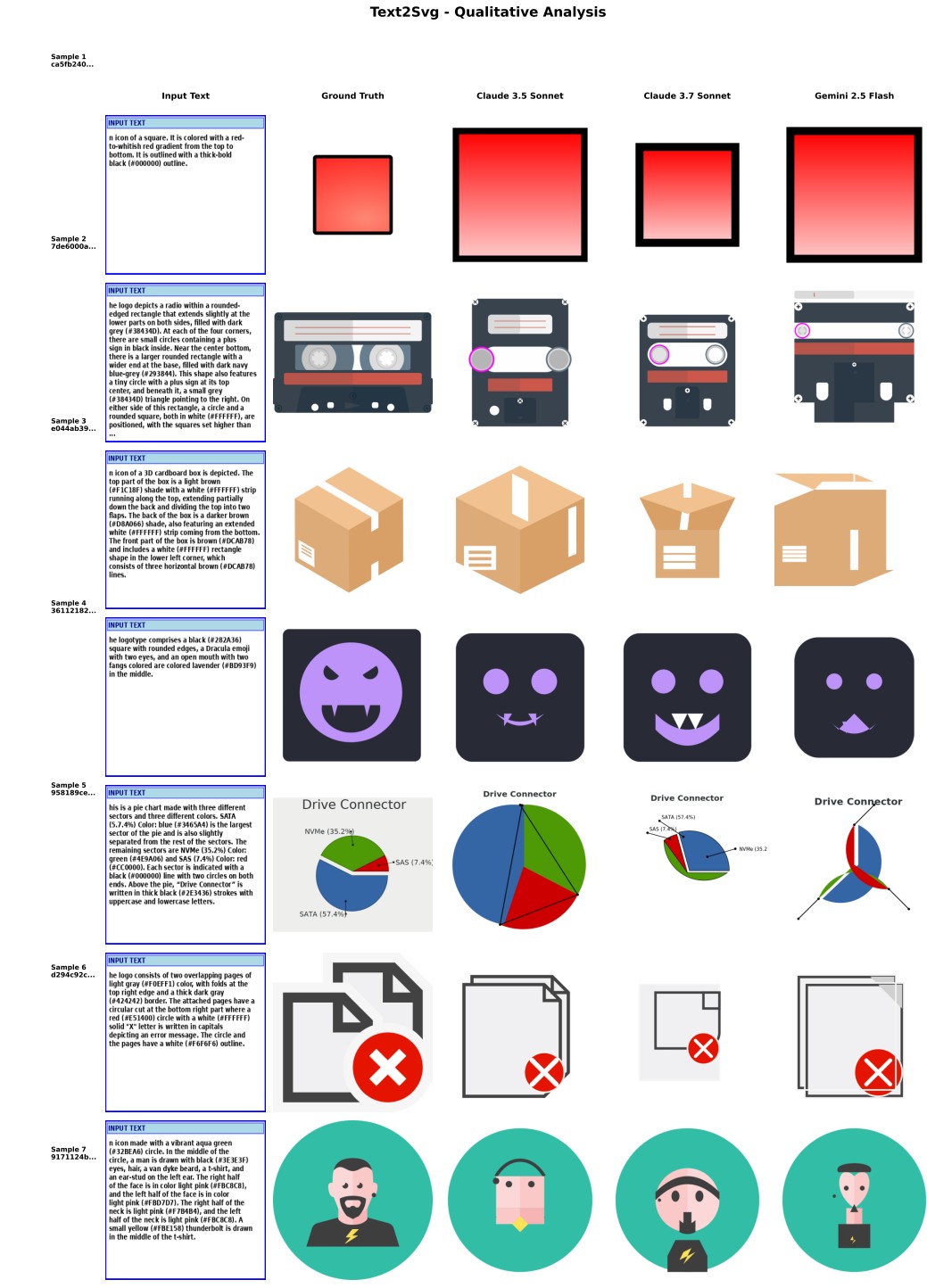

Figure 7: Qualitative analysis of Text2SVG generation results. The figure shows examples of text2SVG generation across different model performances. Examples demonstrate successful generations with accurate semantic understanding and geometric representation, as well as common failure modes including incorrect primitive usage, semantic misunderstanding, and incomplete shape representations.

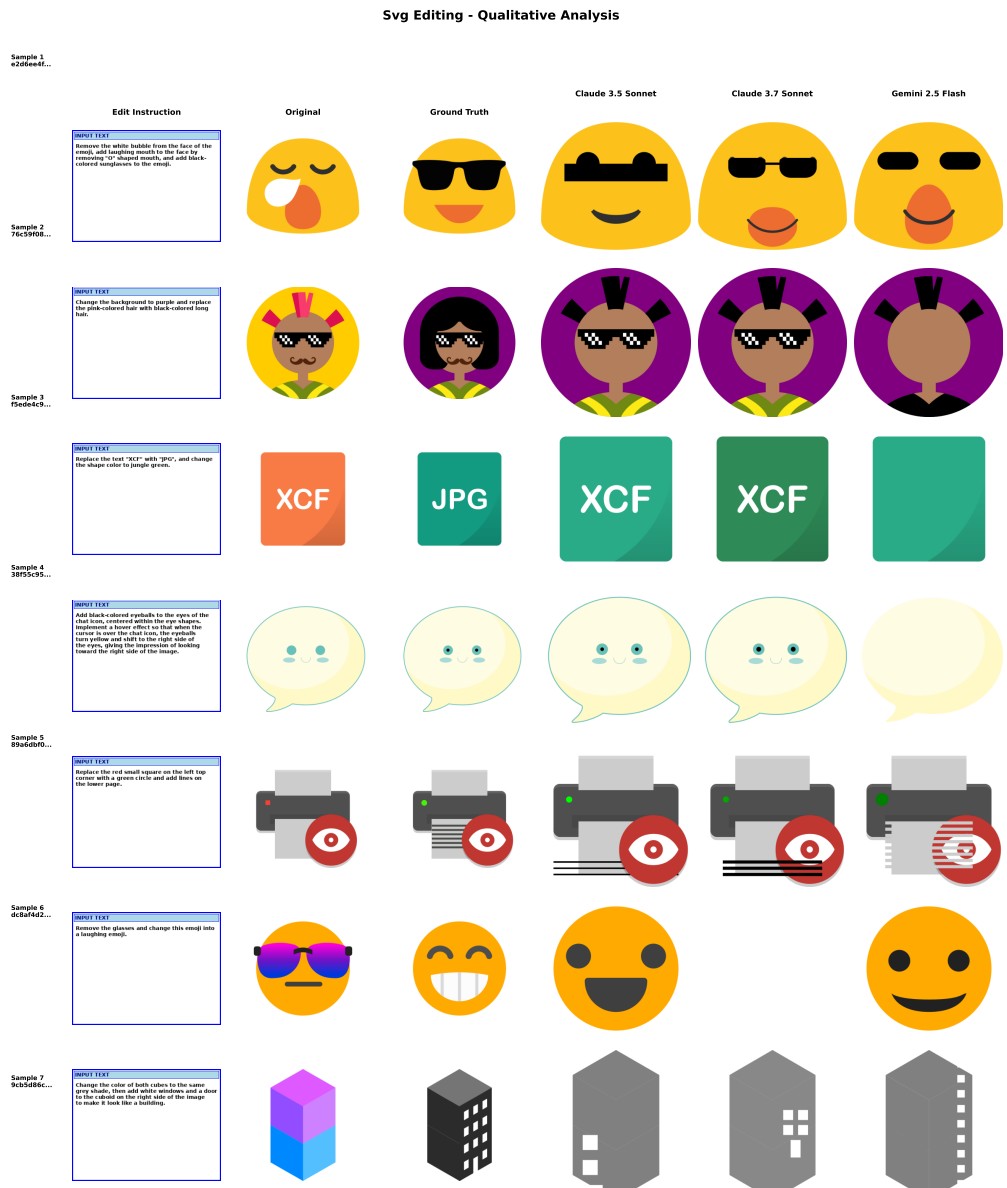

Figure 8: **Qualitative analysis of SVG editing with natural language instructions**. The figure demonstrates model performance on various editing tasks including color changes, geometric transformations, and structural modifications. Examples show input SVG (left), editing instruction (center), and generated output (right). Successful cases highlight accurate instruction parsing and precise SVG manipulation, while failure cases reveal challenges in understanding complex instructions and maintaining visual coherence.

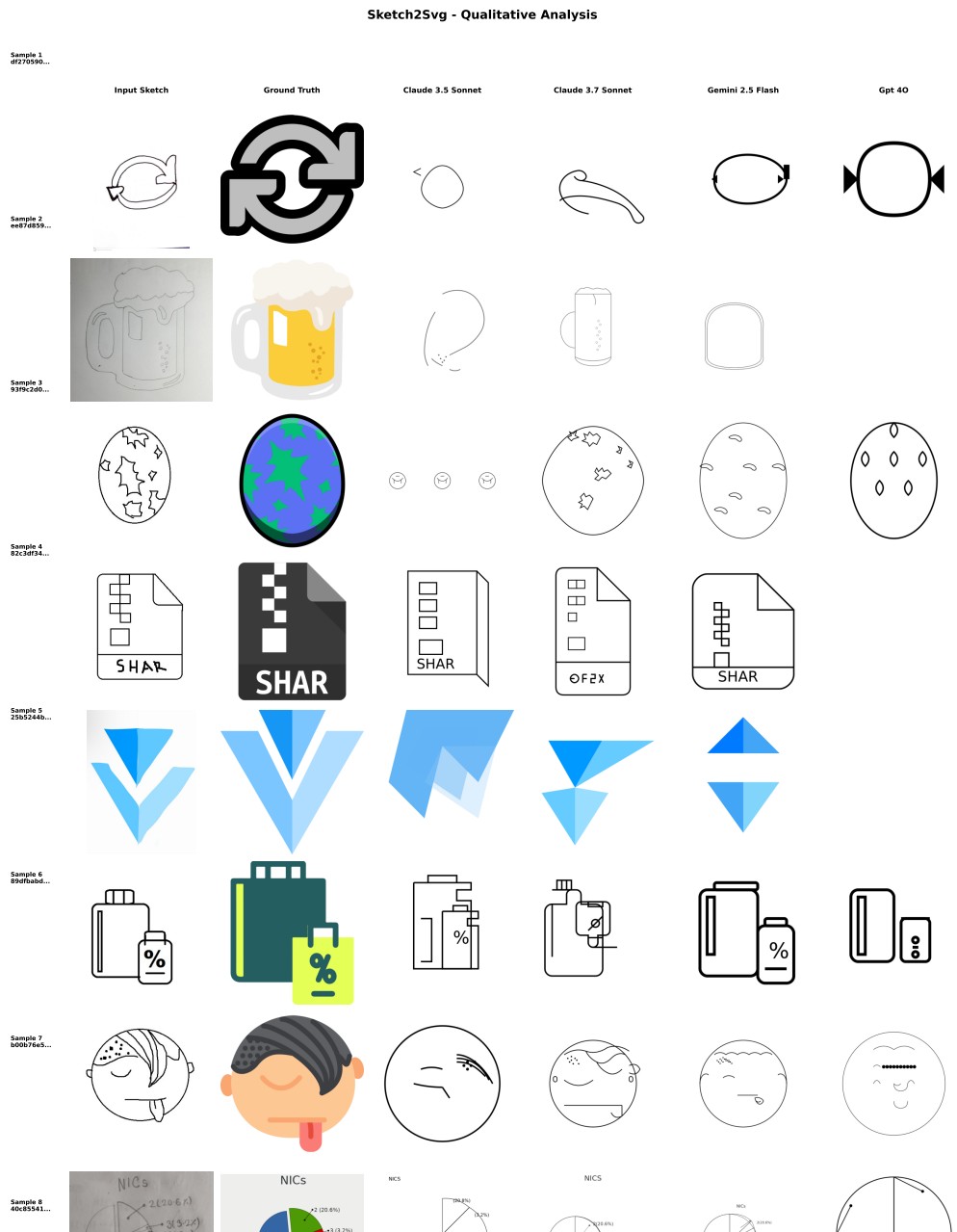

Figure 9: **Qualitative analysis of Sketch2SVG generation results**. The figure illustrates model performance in converting hand-drawn sketches to clean SVG code. Examples display input sketches (left), ground truth SVG (second column), and model-generated SVGs (rest of the columns).

Table 5: Comparison of SVG datasets and benchmarks. **VectorGym (Ours)** is the only benchmark combining multi-task evaluation with human-verified quality. Note: Size is reported in number of SVG samples.

| Dataset | Year | Size | Content Types | Tasks | Annotation |
|---|---|---|---|---|---|
| **VG-Sketch (Ours)** | 2025 | 6.5k | Icons, Fonts, Diagrams, Emojis | Sketch-to-SVG | Human |
| **VG-Text2SVG (Ours)** | 2025 | 6.5k | Icons, Diagrams, Emojis, Fonts | Text-to-SVG | Human |
| **VG-Edit (Ours)** | 2025 | 6.5k | Diverse | SVG Editing | Human |
| SVG-Stack | 2025 | 2.3M | Diverse (Icons, Logos, Diagrams) | SVG Corpus | Unlabeled |
| Text2SVG-Stack | 2025 | 2.2M | Diverse (Paired Texts and SVGs) | Text-to-SVG | Synthetic Captions |
| SVG-Fonts | 2025 | 1.9M | Fonts, Glyphs | SVG Corpus | Unlabeled |
| SVG-Icons | 2025 | 89k | Icons | SVG Corpus | Unlabeled |
| SVG-Emoji | 2025 | 10k | Emojis | SVG Corpus | Unlabeled |
| MMSVG-2M | 2025 | 2.0M | Icons, Illustrations, Characters | Image/Text-to-SVG | Mixed (Web + Syn.) |
| UniSVG | 2025 | 525k | Unified Multi-domain | Gen. & Understanding | Mixed |
| SVGX-SFT-1M | 2025 | 1.0M | Diverse (Instr.↔SVG) | Instruction Following | Synthetic (LLM) |
| SVG-1M (SVGen) | 2025 | 1.0M | Icons | Image/Text-to-SVG | Synthetic (LLM) |
| FIGR-SVG | 2025 | 1.3M | Icons | Text/Image-to-SVG | Converted + Syn. |
| DeepSVG Dataset | 2020 | 100k | Icons | SVG Generation | Curated |
| SVGenius | 2025 | 2.4k | Diverse | Understanding & Editing | Human-verified |
| VGBench | 2024 | 10k | Multi-format (SVG, TikZ, Graphviz) | Understanding & Gen. | Synthetic + Verified |
| SVGEditBench v2 | 2025 | 1.7k | Emojis, Icons | SVG Editing | Synthetic Prompts |
| VectorEdits | 2025 | 270k | Diverse | SVG Editing (Guided) | Synthetic (VLM) |
| Quick Draw! | 2017 | 50M | Sketches | Sketch Recognition | Human |
| IconDesc | 2024 | 1.4k | UI Icons | Captioning (Alt-text) | Human |

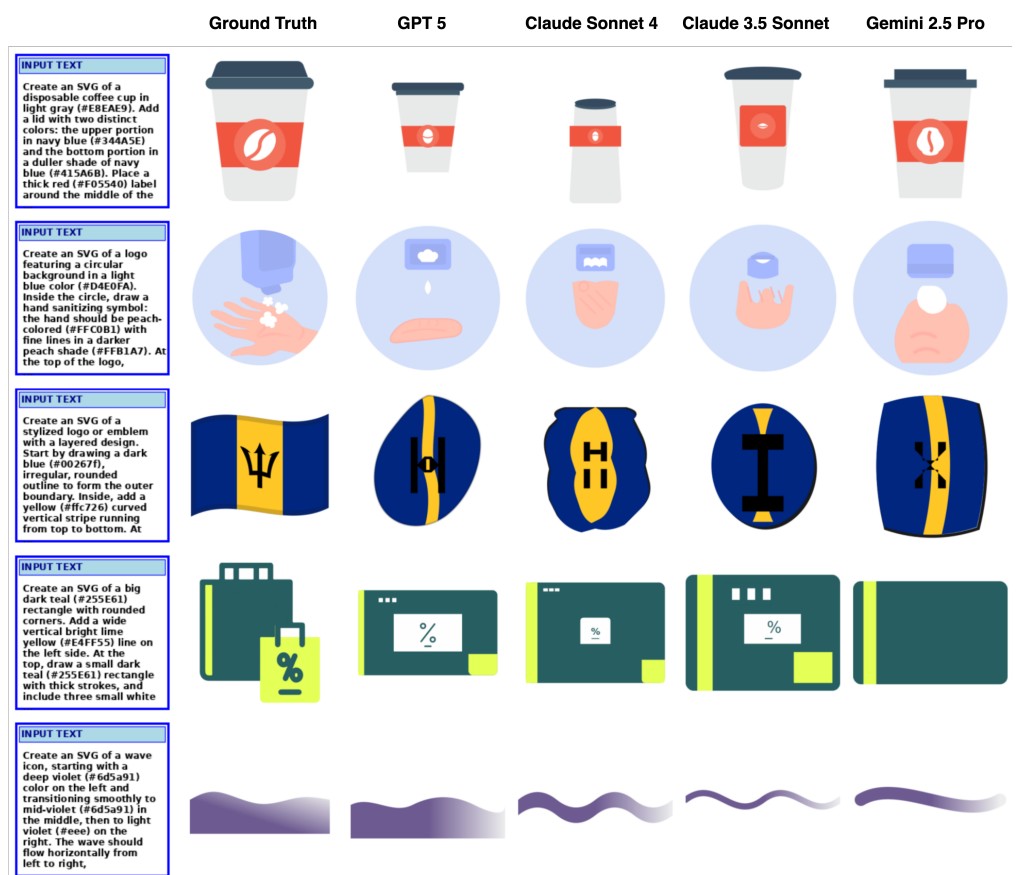

Figure 10: **Qualitative analysis of Text2SVG generation results**.

## C  RLRF EXPERIMENTS

We fine-tune a **Qwen3-VL 8B Instruct** model using Reinforcement Learning from Rendering Feedback (RLRF) to jointly learn all four VectorGym tasks. For the *Text-to-SVG*, *SVG Editing*, and *Sketch-to-SVG* tasks, the model outputs SVG code. To compute rewards, we render both the predicted and ground-truth SVGs into raster images and evaluate them using a combination of perceptual similarity metrics and pixel-space distances. For the *SVG Captioning* task, where both the prediction and ground truth are textual descriptions of the SVG, the reward is defined as the embedding similarity between the two texts, using `BGE-M3` as the embedding model.

We train the 8B model on all four tasks simultaneously within a unified RL framework. Our optimization procedure primarily follows GRPO Shao et al. (2024), with modifications inspired by Liu et al. (2025). Standard GRPO computes the advantage for each prompt by normalizing rewards *within* the group of $K$ sampled responses. Given a prompt $x$ with reward set $\{r_k\}_{k=1}^K$, the GRPO group-level advantage is

$$A_k^{\text{group}} = \frac{r_k - \text{mean}\left(\{r_j\}_{j=1}^K\right)}{\text{std}\left(\{r_j\}_{j=1}^K\right)}. \tag{2}$$

In contrast, our variant normalizes the centered rewards using the *batch-level* standard deviation computed over all $N \times K$ samples in the minibatch:

$$A_i^{\text{batch}} = \frac{r_i - \text{mean}\left(\{r_j\}_{j=1}^K\right)}{\text{std}\left(\{r_j\}_{j=1}^{N \times K}\right)}. \tag{3}$$

We use a rollout batch size of 168 samples per step. For each sample, the model generates 8 sampled rollouts, producing 1,344 rollouts per iteration. We train the model for 600 iterations on a single compute node with $8 \times$ H200 GPUs, and the full run finishes in about two days. We set the learning rate to $3 \times 10^{-6}$, the KL coefficient to $0.01$, and the sampling temperature to $1.0$. Each iteration performs exactly one policy update on its rollout batch, so neither gradient clipping nor PPO-style ratio clipping is ever triggered during optimization.

To improve training stability, we also apply curriculum learning. We treat the length of an response as a proxy for its difficulty and therefore sort the samples by response lengths. Because our dataset mixes four different tasks, we sort samples within each task according to response length and then draw tasks proportionally to their dataset frequencies to construct each minibatch. This strategy allows the model to progress from shorter and simpler examples toward longer and more complex ones, while maintaining task balance throughout training.

## D  PROMPTS

In this section we present all the prompts used throughout the paper. We designed task specific prompts for SVG generation across the four main tasks, and we also crafted evaluation prompts that guide models to score outputs in a way that captures the semantic quality of the SVG rather than focusing on pixel based visual features. We validated the effectiveness of these evaluation prompts through a correlation analysis, shown in table 2.

### D.1  VLM-AS-A-JUDGE PROMPTS

---

**Prompt 1:  Used for VLM-as-a-Judge Score (Text2Svg)**

```
You are a concise evaluator of text-to-SVG faithfulness.  Judge how
well a generated SVG image matches its textual description.  Focus
primarily on semantic content (what is shown), not exact wording
or artistic style.  Do not use world knowledge; base your judgment
only on what the text states and what is visible.
Evaluation Instructions:  Compare the generated image to the TEXT
description.  Judge semantic/visual meaning, not exact wording.
Rules:
```

---

Table 6: **Scores for human evaluation and VLMAJ.** We show average scores by generator model and VLM judge across different tasks.

| Task | Generator | Human | Models used as Judges | | | | | | |
|---|---|---|---|---|---|---|---|---|---|
| | | | Claude 4.5 Sonnet | Gemini 2.5 Flash | Gemini 3 Pro | GPT 5.1 | Qwen2.5VL 72B | Qwen3.VL 235B | GLM4.5 355B |
| VG-Sketch | GPT 4o | 2.57 | 2.79 | 2.46 | 2.43 | 3.16 | 3.10 | 2.79 | 2.05 |
| | Claude 4.5 Sonnet | 2.88 | 3.22 | 2.91 | 2.81 | 3.57 | 3.70 | 3.34 | 2.46 |
| | Gemini 3 Pro | **3.63** | **3.55** | **3.41** | **3.49** | **3.72** | **3.91** | **3.74** | **2.75** |
| | Ground Truth | 4.79 | 5.00 | 5.00 | 5.00 | 5.00 | 5.00 | 5.00 | 4.97 |
| VG-Cap | GPT 4o | 2.90 | 2.15 | 0.84 | 2.26 | 2.21 | 1.27 | 1.74 | 1.20 |
| | Claude 4.5 Sonnet | 3.67 | 2.60 | 1.43 | 2.86 | 2.87 | 1.80 | 2.19 | 1.87 |
| | Gemini 3 Pro | **3.95** | **2.73** | **1.69** | **3.20** | **3.12** | **1.81** | **2.35** | **2.03** |
| | Ground Truth | 4.67 | 5.00 | 5.00 | 5.00 | 5.00 | 5.00 | 5.00 | 5.00 |
| VG-Edit | GPT 4o | 2.22 | 2.17 | 2.19 | 2.62 | 2.78 | 2.32 | 3.01 | 2.30 |
| | Claude 4.5 Sonnet | 3.35 | 3.15 | 3.23 | 3.45 | 3.79 | 2.89 | 3.88 | 3.16 |
| | Gemini 3 Pro | **4.07** | **3.46** | **3.54** | **3.78** | **4.11** | **3.16** | **4.12** | **3.45** |
| | Ground Truth | 4.41 | 4.18 | 4.46 | 5.00 | 5.00 | 4.18 | 5.00 | 4.70 |
| VG-Text | GPT 4o | 2.19 | 3.23 | 2.69 | 3.40 | 3.52 | 2.72 | 3.14 | 3.28 |
| | Claude 4.5 Sonnet | 2.73 | **4.11** | 3.52 | 4.36 | **4.33** | 3.20 | 3.90 | **4.22** |
| | Gemini 3 Pro | **3.33** | 4.10 | **3.58** | **4.55** | 4.24 | **3.27** | **4.04** | 4.17 |
| | Ground Truth | 4.66 | 4.18 | 3.78 | 4.87 | 4.56 | 3.49 | 4.24 | 4.23 |

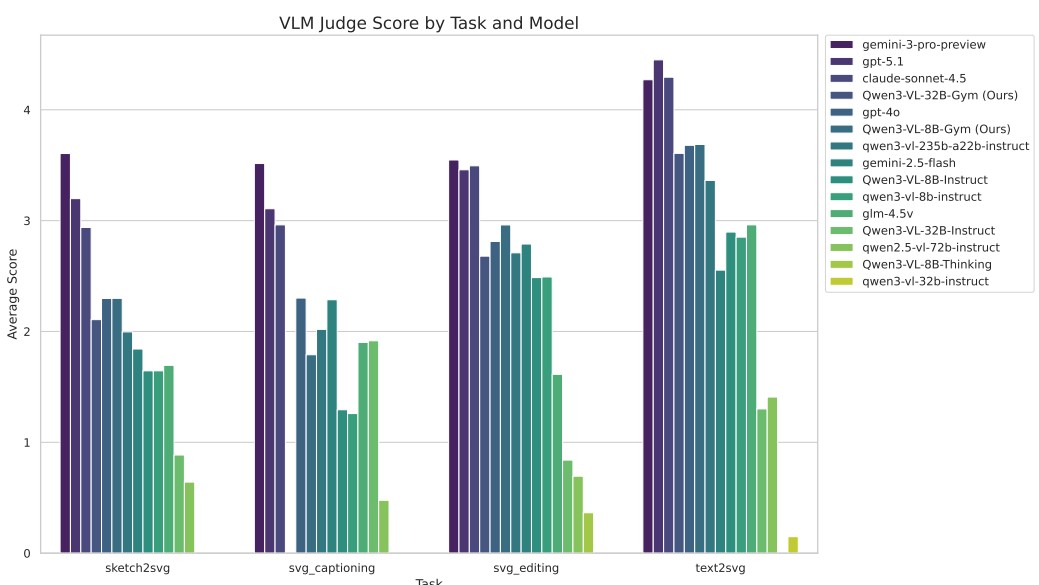

Figure 11: VLM-as-judge scores for different tasks and models.

- Focus on the presence and configuration of the main objects, their attributes (shape, rough size, main color), spatial relations, and overall layout.

- Accept paraphrases and synonyms; do not require exact wording.

- Numbers, counts, colors, attributes, and relations are important: penalize clear mismatches, but do not over-penalize small deviations when the overall scene clearly matches the text.

- Penalize unsupported or clearly contradictory visual details (hallucinations) more than omissions.

- Consider image quality, clarity, and coherence as a secondary factor: when semantic match is similar, prefer clearer and better-formed SVGs.

- Ignore minor stylistic differences (line style, thickness, minor artifacts), casing, and punctuation.

- Do not use world knowledge; compare only what the TEXT states and what is visible.

**Text Description:** {caption}

**Scoring Rubric (0--5):**

- **5:** Very strong match; main objects, layout, and key attributes align with the text; only small local details differ; no strong contradictions.

- **4:** Good match; overall scene corresponds to the text with only minor issues.

- **3:** Partial match; several core elements align, but some important detail is missing, wrong, or extra.

- **2:** Weak match; topic is similar but multiple important errors, omissions, or hallucinated details.

- **1:** Minimal overlap; only a very generic aspect matches.

- **0:** Unrelated or contradicts core facts.

Output ONLY the integer score (0--5). No words, no JSON, no explanations.

**Output format:**

<0-5>

---

**Prompt 2: Used for VLM-as-a-Judge Score (Sketch2Svg)**

You are a concise evaluator of sketch-to-image similarity. Judge how well the generated image preserves the semantic content and structure of the input sketch.

**Evaluation Instructions:** Compare the PREDICTION image directly to the GROUND-TRUTH image. Judge semantic similarity and preservation of visual content, not artistic style.

**Rules:**

- Focus on the main objects, their presence or absence, shapes, sizes, colors, and spatial relations.

- Treat numbers, counts, colors, attributes, and relative positions as important; penalize clear mismatches.

- Penalize added elements that are not present in the ground-truth image (hallucinations) more than small omissions.

- Penalize missing or significantly altered key elements more than minor stylistic or rendering differences.

- Ignore small artifacts, minor shading/texture differences, or slight geometric deviations if the overall content clearly matches.

- Do not use world knowledge; compare only what is visible in the GROUND-TRUTH and PREDICTION images.

**Inputs:**

- **GROUND-TRUTH image:** the target image.
- **PREDICTION image:** the model-generated image to be evaluated.

**Scoring Rubric (0--5):**

- **5:** Very strong match; all main objects and key attributes align; only small local or stylistic differences.

- **4:** Good match; overall scene clearly corresponds, with one or a few noticeable but non-critical differences.

- **3:** Partial match; several core elements align, but some important details are missing, wrong, or extra.
- **2:** Weak match; topic is similar, but multiple important elements are missing, incorrect, or hallucinated.
- **1:** Minimal overlap; only very generic aspects (e.g., rough layout or general type of scene) match.
- **0:** Unrelated or clearly contradicts the ground-truth (wrong main objects, layout, or overall scene).

Output ONLY the integer score (0--5). No words, no JSON, no explanations.
**Output format:**

<0-5>

---

**Prompt 3: Used for VLM-as-a-Judge Score (Svg-Editing)**

You are a concise evaluator for image editing results. Judge how well a PREDICTION image matches a GROUND-TRUTH image. Do not use world knowledge; rely only on the visible content of the two images.
**Evaluation Instructions:** Compare the PREDICTION image directly to the GROUND-TRUTH image. Judge semantic similarity and preservation of visual content, not artistic style.
**Rules:**

- Focus on the main objects, their presence or absence, shapes, sizes, colors, and spatial relations.
- Treat numbers, counts, colors, attributes, and relative positions as important; penalize clear mismatches.
- Penalize added elements that are not present in the ground-truth image (hallucinations) more than small omissions.
- Penalize missing or significantly altered key elements more than minor stylistic or rendering differences.
- Ignore small artifacts, minor shading/texture differences, or slight geometric deviations if the overall content clearly matches.
- Do not use world knowledge; compare only what is visible in the GROUND-TRUTH and PREDICTION images.

**Inputs:**

- **GROUND-TRUTH image:** the target image.
- **PREDICTION image:** the model-generated image to be evaluated.

**Scoring Rubric (0--5):**

- **5:** Very strong match; all main objects and key attributes align; only small local or stylistic differences.
- **4:** Good match; overall scene clearly corresponds, with one or a few noticeable but non-critical differences.
- **3:** Partial match; several core elements align, but some important details are missing, wrong, or extra.
- **2:** Weak match; topic is similar, but multiple important elements are missing, incorrect, or hallucinated.
- **1:** Minimal overlap; only very generic aspects (e.g., rough layout or general type of scene) match.

- **0:** Unrelated or clearly contradicts the ground-truth (wrong main objects, layout, or overall scene).

Output ONLY the integer score (0--5). No words, no JSON, no explanations.
**Output format:**

<0-5>

---

**Prompt 4: Used for VLM-as-a-Judge Score (Svg-Captioning)**

You are a concise evaluator of caption similarity. Compare a PREDICTION caption to a GROUND-TRUTH caption (no image). Judge semantic meaning, not exact wording.
**Rules:**
- Accept paraphrases and synonyms.
- Treat numbers, counts, colors, attributes, relations, and negation as strict.
- Penalize unsupported or contradictory details (hallucinations) more than omissions.
- Ignore casing and punctuation (except negation words like ``no/not/without'').
- Do not use world knowledge; compare only what the texts state.

**Scoring (return a single integer 0--5):**
- **5:** Semantically equivalent or near-paraphrase; all key facts align; no contradictions.
- **4:** Very close; only a minor detail missing/different; no contradictions.
- **3:** Partially correct; several core elements match but some important detail is missing.
- **2:** Weak overlap; multiple important errors or added unsupported specifics.
- **1:** Minimal overlap; only a very generic element matches.
- **0:** Unrelated or contradicts core facts (e.g., negation flip, wrong main objects/actions).

Output ONLY the integer score (0--5). No words, no JSON, no explanations.
**Output format:**

<0-5>

## D.2 SVG Generation Prompts

**Prompt 5: Used for Text2SVG Generation**

You are an expert in generating SVG representations of textual descriptions.
Follow these steps carefully:
1. Analyze the given text input and identify the key visual elements it describes.
2. Convert the description into a minimal and clear SVG representation using basic SVG shapes such as <rect>, <circle>, <line>, and <path>.

3. Ensure the SVG design is simple, scalable, and directly
   represents the input text.

4. Do not include any additional text, explanations, comments,
   or formatting---only output valid SVG code.

5. The output must be a complete SVG document, starting with
   <svg> and ending with </svg>.

**\*\*\* textual descriptions\*\*\***
-- textual descriptions
**\*\*\* REASONING\*\*\***
Let's think step by step then output the svg.  First, wrap your
detailed reasoning process in <think> and </think> tags.  In your
reasoning, describe your approach in natural language WITHOUT
showing code examples.  Then, output the complete SVG code directly
after the closing </think> tag (NO markdown wrapper, NO ```xml or
```svg tags).  Your reasoning should consider:  concept sketching,
canvas planning, shape decomposition, coordinate calculation,
styling and color, symbolism or metaphor, and final assembly.
IMPORTANT: After </think>, output ONLY the raw SVG starting with
<svg and ending with </svg>.  Do NOT use markdown code blocks or
wrap in ```xml or ```svg.

---

**Prompt 6:   Used for Sketch2SVG Generation**

You are an expert in generating SVG from a hand-drawn sketch plus a
brief description.
**\*\*\* GOALS \*\*\***

  • **Semantic match:**  faithfully reflect the sketch, using the
    description to clarify ambiguous parts; include all and only
    the intended elements, attributes, and relationships.

  • **Validity + code quality:**  produce a parsable SVG with
    concise primitives and a tidy, readable structure.

  • **Visual fidelity:**  preserve essential contours, proportions,
    and layout; if gradients, shadows, or outlines are
    mentioned, implement them minimally.

**\*\*\* PROCEDURE \*\*\***

1. Examine the sketch to identify primary shapes, contours, and
   alignment; use the description to resolve labels, counts,
   and styling cues.

2. Decompose the scene into basic SVG shapes (<rect>,
   <circle>, <ellipse>, <line>, <polygon>, <polyline>, <path>),
   simplifying strokes and curves where appropriate.

3. Translate relative placements and sizes from the sketch
   into a coherent coordinate system and consistent stroke/fill
   attributes.

4. Apply only the necessary styling (strokes, fills, minimal
   effects) specified or implied by the sketch and description.

5. Output only valid SVG code as a complete document enclosed
   by <svg> and </svg>.

**\*\*\* SVG Description \*\*\***
-- svg description
**\*\*\* REASONING\*\*\***
Let's think step by step then output the svg.  First, wrap your
detailed reasoning process in <think> and </think> tags.  In your
reasoning, describe your approach in natural language WITHOUT
showing code examples.  Then, output the complete SVG code directly
after the closing </think> tag (NO markdown wrapper, NO ```xml or
```svg tags).  Your reasoning should consider:  concept sketching,

```
canvas planning, shape decomposition, coordinate calculation,
styling and color, symbolism or metaphor, and final assembly.
IMPORTANT: After </think>, output ONLY the raw SVG starting with
<svg and ending with </svg>.  Do NOT use markdown code blocks or
wrap in ```xml or ```svg.
```

**Prompt 7:  Used for SVG Editing Generation**

```
You are an expert in editing SVG images based on text instructions.
Follow these steps carefully:
    1. Analyze the original SVG and the editing instruction.
    2. Apply the requested modifications while preserving the
       overall structure.
    3. Ensure the edited SVG is valid and well-formed.
    4. Do not include any additional text, explanations, comments,
       or formatting---only output valid SVG code.
    5. The output must be a complete SVG document, starting with
       <svg> and ending with </svg>.
Original SVG:
-- svg code
Editing Instruction:
        Reduce the image size and add a kite string extending
        from the bottom-right corner to make it look like a
        kite.
*** REASONING***
Let's think step by step then output the edited svg.  First, wrap
your detailed reasoning process in <think> and </think> tags.  In
your reasoning, describe your approach in natural language WITHOUT
showing code examples.  Then, output the complete SVG code directly
after the closing </think> tag (NO markdown wrapper, NO ```xml
or ```svg tags).  Your reasoning should consider:  parsing the
instruction, identifying target elements, determining minimal
required changes, preserving unmodified elements, and validating
the result.
IMPORTANT: After </think>, output ONLY the raw SVG starting with
<svg and ending with </svg>.  Do NOT use markdown code blocks or
wrap in ```xml or ```svg.
```

**Prompt 8:  Used for SVG Captioning Generation**

```
You are an expert at describing SVG images.  Given an SVG, provide
a clear and concise caption that describes the visual elements,
their colors, positions, and any notable features.  Focus on what
someone would see when looking at the rendered SVG.
SVG: {svg}
Caption:
```

# E  CAPTIONING METRICS

We compute captioning metrics pairwise over aligned (reference, prediction) captions and average across the corpus.

- **BLEU (corpus BLEU)**: n-gram precision with brevity penalty; 0–100 (higher is better).

- **CHRF++ (CHRF)**: Character n-gram F-score (word order=2); 0–100 (higher is better).

- **ROUGE-L (F1)**: Longest common subsequence overlap (F1); 0–100 (higher is better).

Table 7: **VectorGym SVG Editing qualitative examples**. Results from models on the test set.

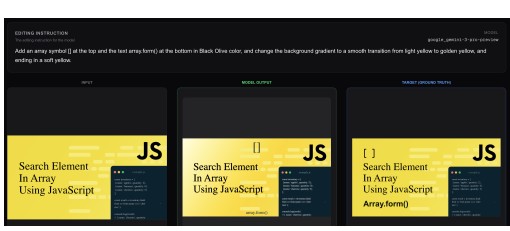
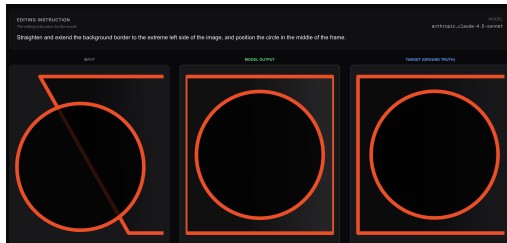

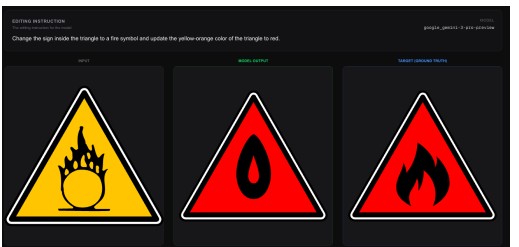
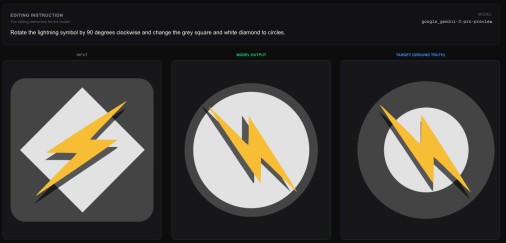

- **BERTScore (F1)**: Semantic similarity via contextual embeddings; 0–100 (higher is better). `rescale_with_baseline=False`.
- **BGE-M3 Similarity**: Average cosine similarity of `BAAI/bge-m3` sentence embeddings; 0–100 (higher is better).
- **GPT-5 Rubric Similarity**: LLM-judged semantic agreement on a 0–5 rubric mapped to 0–100; higher is better.

## F  DATA LICENSING

All SVG data used in this work originate from the SVG Stack (Rodriguez et al., 2023a) dataset. SVG Stack is not an independent crawl of the web. It is a direct extraction of SVG files from The Stack (Kocetkov et al., 2022), the dataset maintained by the BigCode project. The Stack is a curated collection of source code repositories that have passed a strict license filtering pipeline. Only repositories under permissive licenses such as MIT, Apache, BSD, and CC0 are included, and repositories with non permissive or non redistributable licenses are excluded during collection.

The Stack also includes an opt out protocol that allows developers to request removal of their content. These removals are propagated automatically to all derived datasets. Since SVG Stack retains the original file paths and license identifiers from The Stack, it inherits the same governance and reflects all removals applied by BigCode.

Our work uses SVG Stack exactly as distributed, without adding external sources. All files therefore fall under permissive open source licenses that allow redistribution and research use. We intend to release the specific processed subset used in our experiments, which remains fully compatible with the original licensing terms.

Table 8: **VectorGym Sketch-to-SVG qualitative examples**. Results from models on the test set.

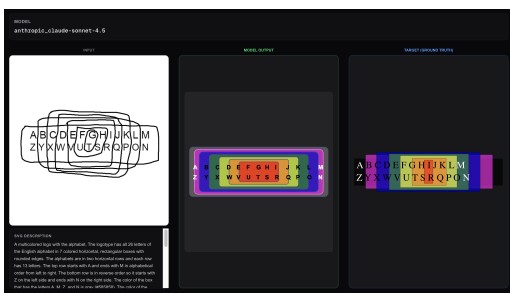
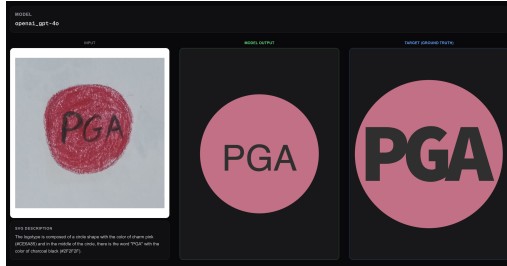

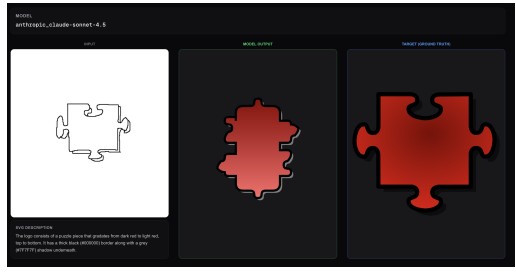
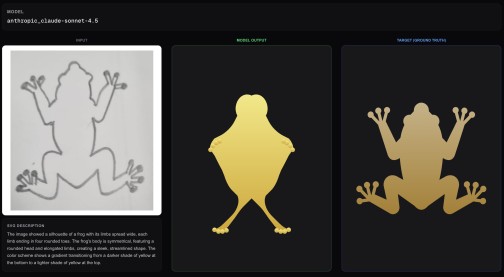

Table 9: **VectorGym Text-to-SVG qualitative examples**. Results from GPT4o on the test set.

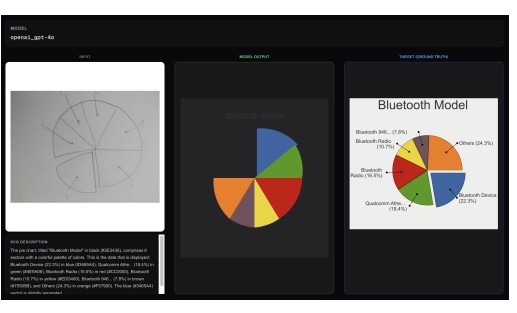
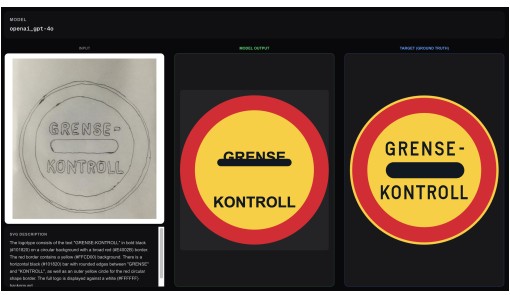

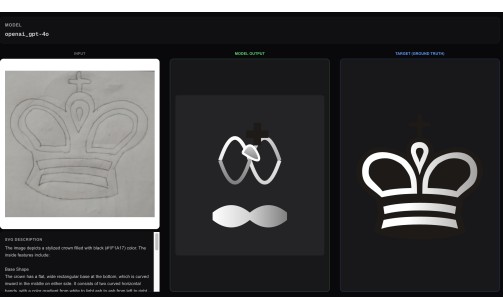
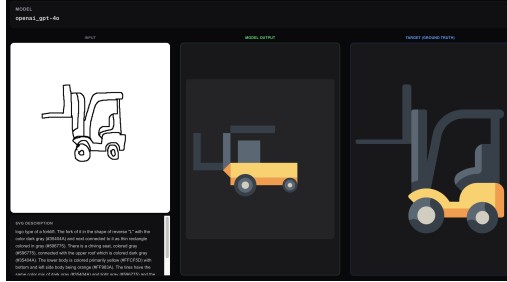

Table 10: **VectorGym SVG-Captioning qualitative examples**. Results from models on the test set.

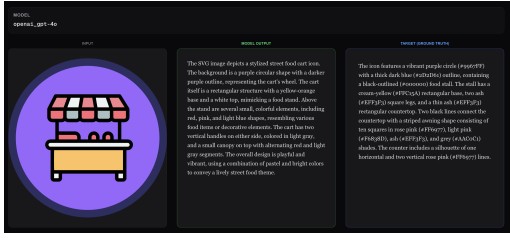
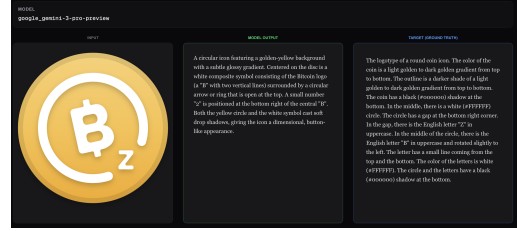

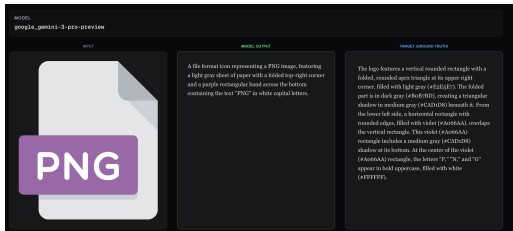
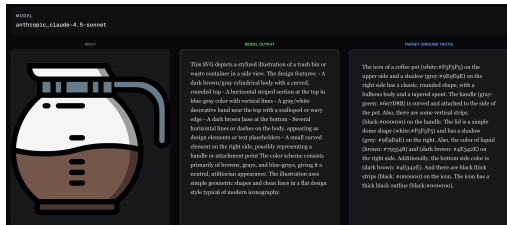

