# OpenReview forum: "VectorGym: A Multi-Task Benchmark for SVG Code Generation and Manipulation"
_ICLR.cc/2026/Conference — Submitted to ICLR 2026_

### Official Review · Reviewer_8J5j · 2025-10-31

**Soundness:** 2
**Presentation:** 2
**Contribution:** 2
**Rating:** 4
**Confidence:** 4

**Summary:**

This paper introduces **VectorGym**, a comprehensive benchmark dataset for evaluating SVG-related tasks.
VectorGym includes multiple tasks such as **sketch-to-SVG conversion, SVG editing, text-to-SVG generation, and SVG captioning**.
The authors evaluate the performance of various **Vision-Language Models (VLMs)** on these datasets.
Furthermore, they assess the results using multiple evaluation metrics, including **VLM-as-a-judge** approaches.

**Strengths:**

- 1. The paper proposes a human-annotated dataset for several **SVG generation tasks**.
- 2. The paper provides a detailed evaluation of the **VLM-as-a-judge** framework.
- 3. Multiple models are evaluated, and the characteristics of each task are analyzed.

**Weaknesses:**

- 1. Although the created dataset is divided into train / validation / test splits for evaluation, only zero-shot evaluation is performed, so the effect of using the data for fine-tuning remains unclear.
- 2. The paper mentions that an LLM was utilized to assist in drafting the Related Work section; however, the citation link for SVGEditBench directs to an entirely unrelated publication, indicating a potential instance of hallucination in the generated text. Moreover, it appears that the description in the paper actually refers to SVGEditBench2, yet in Table 1 it is simply cited as SVGEditBench, which reduces the accuracy and credibility.
- 3. The procedure for constructing the SVG Edit dataset is unclear.
    - 3.1. When humans performed the edits, did they use a drawing tool to modify the images, or did they manually rewrite the SVG code directly?
   - 3.2. Section A.1.4 presents examples of “Required complex edits”, but how were these criteria defined? What other perspectives were considered?
    - 3.3. In SVGEditBench2, edit instructions are generated based on differences between similar images, rather than simple rule-based edits, resulting in more complex instructions. Compared to such existing approaches, does the proposed dataset contain more complex and diverse edit instructions?

**Questions:**

- 1. Regarding the correlation with human evaluation In Table 2.
    - 1.1 why are the results separated between generated samples and ground truth?
    - 1.2. For the SVG edit task, why does the correlation become high for generated samples but drop significantly for ground truth? Does this mean that VLMs assign high scores even when the outputs are correctly generated? If that is the case, wouldn’t it imply that the metric fails to properly distinguish correct generations, thus questioning its reliability as a trustworthy evaluation measure?
    - 1.3 In the Sketch2SVG task, why does the correlation appear lower compared to the other tasks?
- 2. Benchmarks such as SVGenius also provide comprehensive evaluations across multiple tasks, including editing, understanding, and generation. In comparison to such prior benchmarks, what are the main differences or novel contributions of this paper? Are there distinctions beyond the inclusion of sketch-based data?

---

> ### Author Response · Authors · 2025-11-26
>
> **Thanks to the reviewer for their time and effort in evaluating our paper.** We are encouraged that you found our key contributions strong, and **we address the issues and clarifications you raised with the goal of making the work stronger**. Your comments have greatly improved the manuscript.
>
> --------------------------
>
> ## Q1. Lack of Training Experiments.
>
> >Although the created dataset is divided into train / validation / test splits for evaluation, only zero-shot evaluation is performed, so the effect of using the data for fine-tuning remains unclear.
>
> **Thanks for the suggestion, and you are right.** We introduced a training dataset but did not use it in the initial manuscript. **To fully explore this resource and provide meaningful insights, we ran an experiment using the 6.7k training samples** with a base **Qwen3VL 8B model on 8 H200 cards for one week**. This was done through a GRPO based RL training setup, where we applied our carefully designed prompts and inference framework to train the model to perform all four tasks simultaneously, with curated rewards tailored to each task.
>
> The results are now included in Tables 3 and 4, and are also visible in the qualitative examples shown in Figures 3 and 4. For convenience, we have uploaded the samples here: https://imgur.com/a/fXA40SI and https://imgur.com/a/ENaQeXD.
>
> **Results show that using an RL setup to maximize our curated rewards produces a very strong model.** It becomes the **best performer in the open source category** and competes closely with closed models, despite being only 8B in size.
>
> Please see the revised manuscript and Appendix C for additional details.
>
> ----------------------
>
> ## Q2. Problem with Related Work Section
>
> > The paper mentions that an LLM was utilized to assist in drafting the Related Work section; however, the citation link for SVGEditBench directs to an entirely unrelated publication, indicating a potential instance of hallucination in the generated text. Moreover, it appears that the description in the paper actually refers to SVGEditBench2, yet in Table 1 it is simply cited as SVGEditBench, which reduces the accuracy and credibility.
>
> **We sincerely apologize for this issue, and thank you for raising this point.** You are correct, the citation for SVGEditBench was incorrect. This issue originated during a late revision where we used an LLM assisted pass to reorganize the bibliography, which introduced an erroneous entry that we unfortunately did not catch. We appreciate the reviewer bringing this to our attention. We agree with the reviewer that this is a serious flaw, and we take full responsibility for it. We hope this does not significantly affect the credibility of our work.
>
> You are also correct that the textual description referred to the V2 version of the benchmark, while Table 1 listed it simply as SVGEditBench. This inconsistency reduced clarity and accuracy. **We have now corrected both the citation and the naming so that the description and the table refer to the appropriate versions of the benchmark.**
>
> **We re-done the Related Work section and revised the complete bibliography** to ensure that all references are accurate and consistent. The revised manuscript reflects these corrections in red.
>
>
> ----------------------
>
> ## Q3. Clarify SVG Editing Dataset
>
> > Q3.1. When humans performed the edits, did they use a drawing tool to modify the images, or did they manually rewrite the SVG code directly?
>
> **Yes. The annotators were equipped with both drawing tools and code editing tools to handle the full complexity of the tasks**, including adding elements, moving or rotating components, and making structural changes. In many cases they also modified the SVG code manually when this was the most direct way to achieve the intended edit.
>
> All annotators were external professional vendors selected for their expertise in design and coding. They were free to use their own preferred toolsets to complete the challenging SVG editing tasks effectively.

---

> > ### Author Response · Authors · 2025-11-26
> >
> > > Q3.2. Section A.1.4 presents examples of “Required complex edits”, but how were these criteria defined? What other perspectives were considered?
> >
> > In our setup, **complex annotations are human created editing instructions and SVG modifications that require deeper knowledge of SVG syntax, including the use of higher order primitives such as text, gradients, or animations**. These edits also **demand semantic understanding, multi step reasoning, and design intent that cannot be achieved through simple geometric or algorithmic rules**. They include actions like adding new objects, integrating external SVG elements, inserting text with meaningful placement, restructuring layouts, or applying several coordinated edits at once. *These reflect realistic expert design workflows and cannot be reproduced by rule based procedures.*
> >
> > To produce them, annotators with strong SVG expertise used dedicated drawing tools, coding utilities, and curated SVG assets. We have added these details to the revised manuscript.
> >
> > *For example, Figure 1 (right side, https://imgur.com/a/3oZdaGt) illustrates an edit that requires changing a circle into a triangle both visually and textually, showing the type of semantic and structural modifications involved.* **Additional examples of these complexities are provided now in Figure 2 (see updated manuscript).**
> >
> > ----------------------
> >
> > > Q3.3. In SVGEditBench2, edit instructions are generated based on differences between similar images, rather than simple rule-based edits, resulting in more complex instructions. Compared to such existing approaches, does the proposed dataset contain more complex and diverse edit instructions?
> >
> > Great question, thanks for raising it. **The insight in SVGEditBench using that technique is clever, since the edits produced by differencing are indeed more complex than simple algorithmic transforms.** At first, *we tried to apply a similar strategy on our dataset*, which is built from SVG-Stack (more diverse GitHub SVGs), *but it did not work well*. In our case, many SVGs are visually similar while being very different at the code level, so this approach only works reliably when all samples come from the same generator and share a consistent SVG structure.
> >
> > That said, we can confirm that **our edits are substantially more complex**. They require semantic understanding, design intent, and multi step reasoning. **For instance, in Figure 1 (right side, https://imgur.com/a/3oZdaGt), the model must change a circle into a triangle and update the accompanying text, not just apply a local geometric transform.** We have also added a new Figure 2 to the main paper, that showcases several of these more challenging edit tasks.
> >
> >
> > ----------------------
> >
> > ## Q4. Regarding the correlation with human evaluation In Table 2.
> >
> > **We revised the correlation analysis with human evaluation and found several issues caused by too few datapoints.** *Given your feedback, and the other reviewer's, **we decided to rerun the entire study** in a more robust way by **doubling the samples per task from 50 to 100**, **increasing the number of human evaluators from 5 to 17*, and **expanding both the set of VLM judges and the generator models**, notably including **Gemini 3 Pro**. The manuscript now reflects these updates, and the new correlation table together with the **average human and VLM scores can be found here: https://imgur.com/a/VqzzsE5**
> >
> >
> > > Q4.1 why are the results separated between generated samples and ground truth?
> >
> > **We removed the columns related to ground truth** samples and added a dedicated row for the ground truth validation set for each task. This row shows how the VLM judges score the targets, which should ideally be high since these are the correct answers. As mentioned before **we re-ran the entire set of evaluations and human evaluations** with improved VLM judge prompts based on suggestions from other reviewers. **The corrected version is now included in the manuscript, see Table 2 highlighted in red.**

---

> > > ### Author Response · Authors · 2025-11-26
> > >
> > > ----------------------
> > >
> > > > Q4.2. For the SVG edit task, why does the correlation become high for generated samples but drop significantly for ground truth? Does this mean that VLMs assign high scores even when the outputs are correctly generated? If that is the case, wouldn’t it imply that the metric fails to properly distinguish correct generations, thus questioning its reliability as a trustworthy evaluation measure?
> > >
> > > We **revisited the metric** for this task and **confirmed that the metric** used in the submission is **not flawed**. Under standard inspection it behaves as expected. *The drop you observed was due to an imbalance in human evaluation scores for a subset of ground truth samples in the editing task*. After rechecking, we verified that the VLM judge assigns high scores to the ground truth, and the discrepancy came from having too few human ratings on those particular examples. **We have added this expanded analysis in Table 6 (see manuscript, or here: https://imgur.com/a/VqzzsE5)**
> > >
> > > ----------------------
> > >
> > > > Q4..3 In the Sketch2SVG task, why does the correlation appear lower compared to the other tasks?
> > >
> > > Please refer to the previous questions, and our new results.
> > >
> > > ----------------------
> > >
> > >
> > > ## Q5. Differences with other SVG Benchmarks
> > >
> > > > Benchmarks such as SVGenius also provide comprehensive evaluations across multiple tasks, including editing, understanding, and generation. In comparison to such prior benchmarks, what are the main differences or novel contributions of this paper? Are there distinctions beyond the inclusion of sketch-based data?
> > >
> > > **First, we have expanded and clarified the dataset comparison in Table 1 (https://imgur.com/a/tuCBNwh)**
> > >
> > > Our main differences compared with SVGenius are the following:
> > >
> > > **1. More diverse and real world SVG data.**
> > >
> > > Our data is sourced from GitHub, giving a **broad and natural distribution that includes diagrams, icons, logos, illustrations, UI assets,** and many other categories. Prior datasets often focus only on icons or emojis. This also leads to *broader primitive coverage, including text elements, gradients, polygons, groups, and even SVG animations.*
> > >
> > > **2. Complex editing data.**
> > >
> > > Our **editing annotations have the goal to make it challenging to models.** They are created by professional designers and coders who were asked to produce challenging, realistic edits. **These include multi step changes, structural modifications, and design level reasoning, alongside complex higher level primitive understanding.** Previous editing datasets rely mainly on *algorithmic or rule based transformations, which do not capture the complexity of real design workflows.*
> > >
> > > **3. Introduction of sketch based SVG generation.**
> > >
> > > *Sketch2SVG is a new task and was not explored in prior benchmarks.* It adds a new modality and provides a stronger test of multimodal understanding.
> > >
> > > **4. A distinct human centered data pipeline.**
> > >
> > > Unlike SVGenius, our pipeline includes humans who **both draw sketches from the SVGs and perform the edits**, therefore **giving two strong conditioning signals per SVG**. The edits follow instructions that are intentionally complex and cannot be solved with simple geometric transformations or minor adjustments. They require semantic understanding and reflect real design actions, such as adding text, combining SVGs, inserting new elements, or applying several coordinated changes.
> > >
> > > **5. Strengthening two existing tasks: Text2SVG and Captioning**
> > >
> > > **These tasks are not new on their own**, but *we provide high quality annotations and apply them to a richer and more realistic distribution.* The data *has higher structural and semantic complexity than prior benchmarks, and we include human validation to support robust evaluation on generation tasks.*
> > >
> > > **6. VLM-Judge metric.**
> > >
> > > **SVGenius relies on standard metrics and does not incorporate VLM based evaluation.** Our benchmark introduces a tailored VLM judge metric designed for SVG tasks, offering complementary semantic signals across different modalities and producing a more comprehensive evaluation setting.
> > >
> > > ----------------------
> > >
> > > **Thanks a lot for your reviews, which have greatly improved our paper.**

---

### Official Review · Reviewer_uxfE · 2025-10-31

**Soundness:** 3
**Presentation:** 3
**Contribution:** 2
**Rating:** 6
**Confidence:** 4

**Summary:**

This paper introduces a new human annotated multi-task benchmark for SVG code generation and manipulation, covering Sketch2SVG, instruction-guided SVG editing, Text2SVG, and SVG captioning. The authors curate 7,000 real-world SVGs, collect human sketches, and detailed captions, and evaluate a wide range of proprietary and open-source VLMs.

**Strengths:**

1. This paper seems to be the first to combine Sketch, Edit, Text and Captioning tasks, marking for its novelty.
2. This paper provides extensive evaluations of different tasks using multiple open source models covering major evaluation metrics.

**Weaknesses:**

1. The evaluation selects the best of 5 sampled outputs using the same VLM-judge metric. That creates an evaluation bias and may overstate real single-sample performance.
2. The VLM-judge validation uses a validation set of only 50 samples per task to compute Pearson correlations with human annotators, which seems to be too small for robust judge selection, given the diversity of SVGs and edit types.

**Questions:**

1. How sensitive is judge selection to the 50-sample validation set?
2. How do rankings change if you (a) use a single deterministic sample, (b) use median/mean over n samples, or (c) use oracle selection based on human scores?

---

> ### Author Response · Authors · 2025-11-25
>
> We thank the reviewer for their time and efforts, and for the thoughful review and recognizing the novelties of our SVG tasks. We respond to their points and concerns as follows. we habe updated the manuscript with your insights and gerartly improved
>
> ## Q1. Problem with best-of-5 sampling.
>
> > The evaluation selects the best of 5 sampled outputs using the same VLM-judge metric. That creates an evaluation bias and may overstate real single-sample performance.
>
> **We agree that this approach is not standard, and we have removed the best of 5 evaluation procedure.** **We re ran inference in a 1 shot setting and now report the results in Tables 3 and 4 (see revised manuscript)**. We also added new models and baselines following your suggested procedure.
>
> ---------------
>
> ## Q2. How sensitive is judge selection to the 50-sample validation set?
>
> >The VLM-judge validation uses a validation set of only 50 samples per task to compute Pearson correlations with human annotators, which seems to be too small for robust judge selection, given the diversity of SVGs and edit types.
>
> Thanks for pointing this out. **We have redesigned the VLM prompt study and expanded it to 100 examples per task to increase diversity.** We also refined the prompts for the VLM judges based on reviewer feedback and **fully re ran the correlation study**. The updated analysis now includes more human annotators (from 5 to 17) and additional judges (Qwen3VL 32B and 235B, GLM4.5V 108B, and Gemini 3 Pro). **The results are presented in Table 2 of the updated manuscript.**
>
> -------------
> ## Q3. Follow-up on the best-of-n sampling issue.
>
> >How do rankings change if you (a) use a single deterministic sample, (b) use median/mean over n samples, or (c) use oracle selection based on human scores?
>
> We recognize that this approach could be confusing, as noted earlier. **To simplify the evaluation and align with current LLM and VLM practices, we have removed the best of n selection and now sample only once per model**. Thanks for pointing this out.

---

### Official Review · Reviewer_wMeh · 2025-10-31

**Soundness:** 2
**Presentation:** 2
**Contribution:** 2
**Rating:** 6
**Confidence:** 4

**Summary:**

The paper introduces VectorGym, a multi-task benchmark designed to evaluate Vision-Language Models (VLMs) on Scalable Vector Graphics (SVG) generation and manipulation. It spans four tasks: Sketch2SVG, SVG Editing, Text2SVG, and SVG Captioning, supported by a 7k-sample human-annotated dataset. The authors propose a VLM-as-judge metric validated via human correlation and benchmark both proprietary and open-source models, finding GPT-5 and Claude-4 to perform best.

**Strengths:**

- Addresses a real evaluation gap in SVG generation and editing with a well-motivated multi-task design.

- Uses human-authored, complex edits and sketches rather than synthetic data.

- Extensive model coverage with consistent zero-shot evaluation across tasks.

**Weaknesses:**

- VLM-as-judge is not novel; prior works in vision-language evaluation (e.g., LLaVA-Bench, EvalAlign) already employ this approach. Framing it as a key contribution is overstated.
- The term “complex human annotations” is vague and not operationalized--no quantification of complexity, annotator agreement, or examples showing what differentiates them from existing datasets.
- Table 1 is misleading--it should explicitly mark whether prior datasets included any human annotation. The current comparison may overstate novelty.
- All four tasks (Text2SVG, Sketch2SVG, Editing, Captioning) have been studied individually in previous benchmarks; the main novelty is unification, not new task design.
- Sketch2SVG evaluation is questionable: sketches lack full color or geometric precision, but evaluation is done against SVGs which are visually richer. This biases visual-similarity metrics that penalize missing colors or fine details absent in the input.
- Circular evaluation flaw: best-of-n generation and selection uses the same VLM-as-judge for scoring, biasing results.
- Reproducibility concerns: heavy reliance on proprietary APIs (GPT-4o, GPT-5) without open-source judge substitute.
- No statistical testing or confidence intervals on leaderboard results.
- VLLM as a Judge prompt may not be specific enough since score thresholds are defined in ranges: while interpreting results, its hard to make a distinction between a score 7 or 8 since both are supposed to represent "Mostyly accurate and complete, minor issues in detail or quality, clear and visually appealing". Self consistency of these
- Potential dataset contamination: LLMs (Qwen2-VL) used for caption validation may overlap with evaluated models. Human validation of the captions should have been supported.

**Questions:**

- How are “complex human annotations” defined and measured?
- In Table 2: Text2SVG and SVG Edit: Claude 4 Sonnet and Gpt4o have the exact same values upto 3 decimal places, as reported. Are you sure these are valid and not a mistake?
- Following up from the previous question, can the authors provide variance or statistical significance for differences <0.2 in judge scores?
-What exact filtering thresholds (token count, color entropy) were applied during curation?
- Can you include examples of human validation vs LLMaaJ in the appendix? Also attach metrics with all qualitative examples to give an idea of how the metrics are significant.

Minor concerns:
Stary characters like upside down ? in line: 385

---

> ### Author Response · Authors · 2025-11-25
>
> **We sincerely thank the reviewer for the constructive feedback and comprehensive comments. Below we address each point and clarify the concerns raised, and we have incorporated these updates into the manuscript.**
>
> -------------
>
> ## Q1. VLM-as-judge is not novel;
> >Prior works in vision-language evaluation (e.g., LLaVA-Bench, EvalAlign) already employ this approach. Framing it as a key contribution is overstated.
>
> **We have toned down the contribution to read “We design a VLM as a judge SVG evaluation metric,” rather than suggesting broader novelty.** We believe this remains an important contribution. **While the idea of using VLMs as judges is not new, our work develops a version tailored to the SVG visual code generation setting and, in particular, to several key SVG use cases**. This provides an evaluation signal that aligns more closely with human criteria and can also support future model improvement through RL based rewards. The design of this metric requires substantial effort and careful validation, and we view the resulting evaluator as a valuable resource for the SVG community.
>
> -----------------------
>
> ## Q2. The term “complex human annotations” is vague and not operationalized
> > No quantification of complexity, annotator agreement, or examples showing what differentiates them from existing datasets.
>
> Thanks for pointing this out. **In our setup, complex annotations are human created editing instructions and SVG modifications that require deeper knowledge of SVG syntax**, including the use of **higher order primitives such as text, gradients, or animations.** These **edits also demand semantic understanding, multi step reasoning, and design intent** that cannot be achieved through simple geometric or algorithmic rules. They include actions like adding new objects, integrating external SVG elements, inserting text with meaningful placement, restructuring layouts, or applying several coordinated edits at once. These reflect realistic expert design workflows and cannot be reproduced by rule based procedures.
> To produce them, annotators with strong SVG expertise used dedicated drawing tools, coding utilities, and curated SVG assets.
>
> **We have added these details to the revised manuscript.**
>
> ------------------------
>
> ## Q3. Table 1 is misleading
> > It should explicitly mark whether prior datasets included any human annotation. The current comparison may overstate novelty.
>
> **We have updated Table 1 to provide a clearer comparison with existing benchmarks.** It is highlighted in red in the edited manuscript ^^(see the new table here https://imgur.com/a/tuCBNwh)^^. The revised table emphasizes the core strengths of our setup:
>
> 1. The **SVG data is diverse and real world-like**, since it is sourced from GitHub. It includes diagrams, icons, logos, illustrations, and many other categories, while several prior datasets focus only on icons or emojis.
>
> 2. **SVG Primitive coverage is broader**. Since we do not filter or simplify SVGs, our dataset preserves rich structures that go well beyond simple paths, **including texts, color gradients, and animations**, among others.
>
> 3. Similar to the strongest existing benchmarks such as VGBench, UniSVG, and SVGenius, **our setup spans multiple tasks including generation and understanding**.
>
> 4. For **editing**, it **complements the excellent work of SVGEditBench V1 and V2 and SVGenius**, while introducing *edits that are intentionally more difficult and involve multi step design reasoning* and new types of transformation.
>
> 5. It is the **only benchmark that includes Sketch2SVG**, making this a *novel task in the field*.
>
> 6. **All annotations are produced by expert humans with challenging intent**. These complex annotations, described in detail below, were not available in previous datasets and introduce more realistic and semantically meaningful targets.

---

> > ### Author Response · Authors · 2025-11-25
> >
> > ------------------
> >
> > ## Q4. All four tasks have been studied individually in previous benchmarks;
> > > the main novelty is unification  (Text2SVG, Sketch2SVG, Editing, Captioning), not new task design.
> >
> > We respectfully disagree with the statement that all four tasks have been previously established. *Sketch2SVG is, to the best of our knowledge, a new task. Existing SVG benchmarks do not contain human drawn sketches paired with SVG targets.** Our dataset provides the first such collection, **created by human annotators who draw sketches from real SVGs**, which enables a level of abstraction and variability not present in prior work.
> >
> > **Regarding SVG Editing, our task also introduces novelty.** While editing has appeared in **earlier benchmarks**, those settings **rely on synthetic or algorithmically generated transformations. In contrast, our edits are produced by human experts** following instructions specifically **designed to create complex, semantically meaningful modifications that cannot be reproduced by simple geometric or procedural transformations**. Furthermore, our dataset includes higher order **SVG primitives which were not considered in prior work (texts, gradients, animations), making our benchmark more challenging.** Prior benchmarks rely mostly on icon datasets or emojis, so this real world distribution was not previously available for these tasks and makes the evaluations more challenging and representative.
> >
> > We have included a **clear description of what we mean by this “complex annotations”.** We have also included the instructions given to annotators. **Complex annotations in our setup are expert created editing instructions and SVG modifications that require the use of SVG primitives like texts, gradients or animations** and require **semantic understanding** and **multi step design reasoning**. They *involve realistic actions such as adding meaningful content, restructuring layouts, or performing coordinated edits* that cannot be produced through simple geometric rules or algorithmic transformations.
> >
> > We agree that Text2SVG and SVG Captioning have been studied before, but our contribution here is an improved formulation grounded in high quality human validated annotations. These tasks benefit from more realistic distributions compared with previous synthetic datasets, which often lead to trivial edits or limited semantic depth.
> > Overall, the contribution is not only unification, but also the introduction of two tasks with novel data (Sketch2SVG and human curated complex Editing), and improved versions of existing tasks built from more challenging, real world SVG content.
> >
> > ------------------------
> >
> > ## Q5. Sketch2SVG evaluation is questionable
> >
> > >Sketches lack full color or geometric precision, but evaluation is done against SVGs which are visually richer. This biases visual-similarity metrics that penalize missing colors or fine details absent in the input.
> >
> > **We appreciate this observation and agree with it.** The issue arose because our original prompt did not specify whether the output should be colored, which **led models to produce black and white SVGs that resembled the sketches**. To address this, **we have re run the complete Sketch2SVG benchmark with an improved prompt that clearly requests a colored output**. The updated results are now shown in **Figures 3 and 4 (a) (see manuscript)**.
> >
> > We also share them here: https://imgur.com/a/fXA40SI, https://imgur.com/a/ENaQeXD
> >
> > -----------------------
> >
> > ## Q6. Circular evaluation flaw: best-of-n generation and selection uses the same VLM-as-judge for scoring, biasing results.
> >
> > **We have removed the best of 5 evaluation procedure and just run inference on 1-shot**, and report the result on the test set.
> >
> > **We have re-run all the experiments with your proposal** and we also have included the results of the new VLM metric. **Please see the updated section marked in red in the revised manuscript.**).
> >
> > ----------------------
> >
> > ## Q7. Reproducibility concerns: heavy reliance on proprietary APIs (GPT-4o, GPT-5) without open-source judge substitute.
> >
> > **This is a great point and we agree.** To address it, **we have added three open source judges: Qwen2.5VL 72B, Qwen3.VL 235B, and GLM4.5 355B**. We also **re ran the full human evaluation and correlation analysis, following your question and similar feedback from other reviewers.** The updated results are more stable and show that Gemini 3 is the strongest judge, while **the open source Qwen3VL 235B is an excellent replacement**.
> >
> > The new results are visible in Table 2 of the revised manuscript (https://imgur.com/a/VqzzsE5)
> > Please see the updated sections marked in red in the revised manuscript.

---

> ### Author Response · Authors · 2025-11-25
>
> ## Q8. No statistical testing or confidence intervals
> >No statistical testing or confidence intervals are provided on leaderboard results.
>
> It is not standard practice in LLM or VLM leaderboards to report statistical tests or confidence intervals, and most existing evaluation setups do not include them. Running such tests would require a very large number of repeated generations per model, which is not cost effective given the scale of these experiments. The sample sizes we use are already large, and increasing them enough to support reliable statistical testing would multiply the compute cost significantly.
>
> If the reviewer has a specific test in mind, we would be happy to consider it, but in the current landscape of LLM benchmarking this type of statistical analysis is still quite rare.
>
> -----------------------------------
>
> ## Q9. VLLM as a Judge prompt may not be specific enough.
>
> > The VLM judge prompt may be too coarse because score ranges overlap. It is difficult to distinguish between adjacent scores like 7 and 8, since both descriptions are nearly identical, raising concerns about self-consistency.
>
> **The reviewer is correct that the initial VLM as a Judge prompt allowed some freedom within each scoring bracket**, which could make distinctions like 7 vs 8 less clear in practice. To address this, **we redesigned the prompt to enforce much sharper and more interpretable boundaries**. Specifically, *we moved from a 0 to 10 scale with broad ranges to a tighter 0 to 5 scale* where each score has a precise, non overlapping definition. This removes the ambiguity between adjacent scores and improves self consistency.
>
> Along with the other reviewer suggestions to strengthen the analysis, **we reran the full evaluation using the improved prompt, increasing the number of samples, human annotators, and VLM judges**. With this updated setup, the correlations with human ratings are noticeably more stable, and the noise observed in the original VLM judge design is significantly reduced. Please see the updated Table 2.
>
> --------------------
>
> ## Q9. Potential dataset contamination
> > LLMs (Qwen2-VL) used for caption validation may overlap with evaluated models. Human validation of the captions should have been supported.
>
> The concern about possible contamination from using Qwen2 VL for caption validation is understandable, but the risk is very low in our setup. This practice is common, and useful to remove low quality captions. Further, we do not observe any self favouring behaviour in practice. Most importantly, **all captions are independently validated by humans, as described in Appendix A.1.4.**
>
> We report a **full human evaluation in Table 6 (https://imgur.com/a/j5V9CQl)** of the revised manuscript, which **shows no issues with Qwen2.5VL when scoring captions and confirms that humans rate the ground truth data highly**. Overall, the filtering and human validation ensure that the caption set is stable and not affected by model overlap.
>
> ------------------
>
> ## Q10. Complex Human Annotations.
> >How are “complex human annotations” defined and measured?
>
> In our setup, **complex annotations are human created editing instructions and SVG modifications that require deeper knowledge of SVG syntax, including the use of higher order primitives such as text, gradients, or animations.** These **edits also demand semantic understanding, multi step reasoning, and design intent** that cannot be achieved through simple geometric or algorithmic rules. They include actions like adding new objects, integrating external SVG elements, inserting text with meaningful placement, restructuring layouts, or applying several coordinated edits at once. These reflect realistic expert design workflows and cannot be reproduced by rule based procedures.
>
> To produce them, **annotators with strong SVG expertise used dedicated drawing tools, coding utilities, and curated SVG assets**. We have added these details to the revised manuscript.
>
> *For example, Figure 1 (right side, https://imgur.com/a/3oZdaGt) illustrates an edit that requires changing a circle into a triangle both visually and textually, showing the type of semantic and structural modifications involved.* Additional examples of these complexities are provided now in Figure 2.
>
> ------------------
>
> ## Q11. Potential Errors in Results Tables
> > In Table 2: Text2SVG and SVG Edit: Claude 4 Sonnet and Gpt4o have the exact same values upto 3 decimal places, as reported. Are you sure these are valid and not a mistake?
>
> To ensure correctness, **we recomputed all scores and expanded the evaluation to include additional models such as Gemini 3 Pro*, as well as a new result from training a model with GRPO**
>
> Results are more stable now, and we have re-structured the tables to make them more clear. **Please see Tables 3 and 4 in the updated manuscript.**

---

> > ### Author Response · Authors · 2025-11-25
> >
> > ----------
> >
> > ## Q12. Following up on Results Tables.
> > > From the previous question, can the authors provide variance or statistical significance for differences <0.2 in judge scores? -What exact filtering thresholds (token count, color entropy) were applied during curation?
> >
> > As mentioned above, providing statistical significance is not standard practice in current benchmarks, and most existing evaluation setups do not report confidence intervals or significance tests.
> >
> > For the filtering thresholds, we used a **token count range of 2k to 8k** and a **normalized color entropy threshold greater than 0.55**. These **details have now been added to Section 3.2** in the revised manuscript.
> >
> > ---------------
> >
> > ## Q13. Qualitative Examples of Human Validation
> > > Can you include examples of human validation vs LLMaaJ in the appendix? Also attach metrics with all qualitative examples to give an idea of how the metrics are significant.
> >
> > This would be a great addition to our paper. **We now include side by side comparisons of human validation and VLM as Judge on challenging randomly sampled cases, along with the corresponding metric values to illustrate how the scores align.**
> >
> > Figure 4 (https://imgur.com/a/fXA40SI) shows examples with the VLMAJ score displayed on top of each image.
> >
> > Figure 3 (https://imgur.com/a/ENaQeXD) provides additional comparisons across different tasks, including both human and VLM judge scores, highlighting the consistency between them.
> >
> > -----------
> >
> > ## Q14. Minor concerns:
> > >Stary characters like upside down ? in line: 385
> >
> > Thanks, this is now fixed
> >
> > -----------
> >
> > **Thank you very much for the thoughtful and careful review, which has greatly improved our manuscript.**

---

### Official Review · Reviewer_3jQc · 2025-11-02

**Soundness:** 2
**Presentation:** 1
**Contribution:** 2
**Rating:** 2
**Confidence:** 3

**Summary:**

This paper constructs various benchmarks to evaluate the ability of VLMs to understand SVG data and reports their performance across different models. Specifically, the benchmarks are organized into four tasks: Sketch (creating SVGs from drawings), Edit (editing SVGs), Text (generating SVGs from text), and Cap (captioning SVGs). Evaluations are reported for both open-source and proprietary models.

**Strengths:**

The purpose of this paper is clear. It is meaningful to investigate how well LLMs/VLMs can understand the SVG data format. The authors specifically constructed a large-scale dataset and conducted this investigation.

It's also worth noting that various experiments were conducted and evaluations performed for both open-source and proprietary models.

**Weaknesses:**

This paper has the following weaknesses.

## Comparison with Previous Benchmarks

The paper lacks discussion on how it differs from the various benchmarks shown in Table 1. As indicated in Table 1, several benchmarks already exist for evaluating SVG understanding. However, the paper does not discuss what distinguishes its benchmarks from the existing ones, nor does it compare its conclusions and qualitative discussions with those of prior benchmark studies. The absence of such comparisons is problematic.

In the rightmost column of Table 1, previous studies are marked as lacking "Complex Human Annotations," but the meaning of "Complex" is not explained. It is unclear in what sense the annotations in this paper are considered "complex" compared to those in previous works.

## Citation of Nonexistent Papers

Among the papers listed in Table 1, there are significant bibliographic errors in two of them:

- VGBench is cited in this paper as the Findings of EMNLP 2024 by Xia+, which is incorrect. The correct citation is the main conference paper at EMNLP 2024 by Zou+, with a different title. https://aclanthology.org/2024.emnlp-main.213/
- SVGEditBench is attributed to Shu+ in this paper, which is also incorrect. The correct authors are Nishina+, with a different title and arXiv link. https://arxiv.org/abs/2502.19453

In particular, for SVGEditBench, "Changyue Shu" is listed as the author. However, no such person exists, according to a Google search. Therefore, this error is more than a simple BibTeX mistake.

Misidentifying the most relevant comparative works is a serious flaw. It suggests that the authors may not have properly reviewed prior research, despite claiming that earlier works lacked "Complex Human Annotations."

## Insufficient Description of the Proposed Method

The section describing the proposed method (Sec. 3) is only one page long, and everything from page 4 onward is devoted to experiments. There are no details about how the task definitions were established or how the dataset was constructed. While the length of a section does not determine a paper's value, the lack of methodological detail makes the paper feel more like a technical report than a scientific research paper.

## Legal Issues Regarding SVG Data

The authors do not address the legal issues surrounding the SVG data. In Section 3.2, it is stated that SVG data were obtained from the SVG-Stack dataset, and in Section 6 ("Ethics Statement"), it is claimed that licensing was handled appropriately. However, there is no explanation of how this was done. Are the SVGs used distributed under licenses such as MIT or CC that allow redistribution? Moreover, do the authors intend to release the dataset? These points are insufficiently discussed.

**Questions:**

Any comments on legal issues?

**Details Of Ethics Concerns:**

See "Legal Issues Regarding SVG Data" in the weaknesses section.

---

> ### Author Response · Authors · 2025-11-25
>
> We sincerely thank the reviewer for the thoughtful comments and constructive feedback. Below we address each point and clarify the concerns raised, and we have incorporated these updates into the manuscript.
>
> ---------------
>
> ## Q1. Comparison with Previous Benchmarks:
>
> >The paper does not clearly explain how its benchmark differs from those in Table 1 or how its findings relate to prior work. It also labels its annotations as “complex” without defining what this means or how they compare to annotations in earlier benchmarks.
>
> **We have updated Table 1 in the manuscript** (https://imgur.com/a/tuCBNwh) to provide a clearer comparison with existing benchmarks. It is highlighted in red in the edited manuscript .
>
> The revised table emphasizes the core strengths of our setup:
>
> 1. The **SVG data is diverse and real world-like**, since it is sourced from GitHub. It includes diagrams, icons, logos, illustrations, and many other categories, while several prior datasets focus only on icons or emojis.
>
> 2. **SVG Primitive coverage is broader**. Since we do not filter or simplify SVGs, our dataset preserves rich structures that go well beyond simple paths, **including texts, color gradients, and animations**, among others.
>
> 3. Similar to the strongest existing benchmarks such as VGBench, UniSVG, and SVGenius, **our setup spans multiple tasks including generation and understanding**.
>
> 4. For **editing**, it **complements the excellent work of SVGEditBench V1 and V2 and SVGenius**, while introducing *edits that are intentionally more difficult and involve multi step design reasoning* and new types of transformation.
>
> 5. It is the **only benchmark that includes Sketch2SVG**, making this a *novel task in the field*.
>
> 6. **All annotations are produced by expert humans with challenging intent**. These complex annotations, described in detail below, were not available in previous datasets and introduce more realistic and semantically meaningful targets.
>
> The following is the definition of complex human annotations:
>
> **Complex annotations**: In our setup, complex annotations are *human created editing instructions and SVG modifications that require deeper knowledge of SVG syntax**, including the use of higher order primitives such as **text, gradients, or animations**. These edits also demand *semantic understanding, multi step reasoning, and design intent* that cannot be achieved through simple geometric or algorithmic rules. They include actions like adding new objects, integrating external SVG elements, inserting text with meaningful placement, restructuring layouts, or applying several coordinated edits at once. These reflect realistic expert design workflows and cannot be reproduced by rule based procedures.
>
> -----------
>
> ## Q2. Citation of Nonexistent Papers
>
> >Two benchmarks in Table 1 are mis-cited, including one with a non-existent author. This raises concerns about the care taken when reviewing related work.
>
> **We sincerely apologize for this issue.** We want to clarify that the paper was not written from scratch using an LLM. After reviewing the source of the error, **we found that it was introduced during a late pass intended to shorten the Related Work section**, which we had presented in the submission. That pass introduced duplicated entries and replaced correct references with hallucinated ones. We agree with the reviewer that this is a serious flaw, and we take full responsibility for it. We hope this does not significantly affect the credibility of our work.
>
> **We have now rewritten the Related Work section from the ground up**. All citations have been corrected, including the entries for VGBench and SVGEditBench, and we added more precise descriptions of the relevant literature. This error does not reflect a misunderstanding of prior work. Our actual analysis was based on the correct papers, but the faulty BibTeX entries caused confusion in the submitted version.
>
> **We also used this opportunity to review the entire bibliography and ensure that no other issues remain**. After these corrections, we confirm that the revised manuscript accurately covers all relevant research.
>
> **Thanks to your suggestions, we improved the comparison with existing benchmarks (Table 1)  and clarified the notion of complex human annotations.** These refer to human created editing instructions and corresponding SVG modifications that require semantic understanding, multi step reasoning, and design intent, as illustrated in the example in Figure 1: https://imgur.com/a/3oZdaGt
>
> All changes are highlighted in red in the updated manuscript.

---

> > ### Author Response · Authors · 2025-11-25
> >
> > ## Q3. Insufficient Description of the Proposed Method
> >
> > >The section describing the proposed method is very brief, and the paper gives almost no detail on how the tasks or dataset were defined. This lack of methodological explanation makes the work read more like a technical report than a research paper.
> >
> > This section introduces and describes our proposed benchmark. **We presented the task definitions in Section 3.1 and the dataset construction details in Section 3.2 and Appendix A**. Following your suggestion, **we have expanded the task descriptions and added further details on the dataset creation process**. We have also *reorganized Section 3 so that it now provides the full benchmark specification*, including the **evaluation metrics and the design of the VLM as Judge metric**, which were previously only described in the experiments section.
> >
> > At submission time we did not propose a specific training method and instead relied on open and closed VLMs with carefully designed prompting for inference, as described in Section 4.1. **In the revised version we now also include a method based on training a Qwen3VL 8B Instruct model using RL GRPO**, which has been added to Section 4.1 alongside the other baseline methods. *We are happy to further expand Section 3 if you think additional details would be helpful.*
> >
> >
> > ------------------------------
> >
> > ## Q4. Legal Issues Regarding SVG Data
> >
> > >The paper does not explain how licensing for the SVG data was handled. It states that data came from SVG-Stack and that licensing was addressed, but it does not specify what licenses apply or whether redistribution is allowed. It is also unclear whether the authors plan to release the dataset.
> >
> > As explained in the Ethics Statement, our dataset follows good and permissive licenses, therefore no legal issues are anticipated. We clarify below how licensing was handled for all SVG data.
> >
> > Quick answers:
> >
> > **1. Redistribution is allowed**
> > **2. We will release the dataset**
> >
> > Details on licenses:
> >
> > **All SVG files used in our work come from the SVG Stack dataset**. SVG Stack is not an independent crawl. It is a direct extraction of SVG files from BigCode’s The Stack dataset. **The Stack is a curated collection of repositories that meet BigCode’s license restrictions.** Only projects under permissive open source licenses such as MIT, Apache, BSD, and CC0 are included. **Repositories with restrictive or non redistributable licenses are excluded by design.**
> >
> > Since **SVG Stack** is a direct subset of The Stack, it **fully inherits its permissive licensing status and its governance.** This is why we stated in the ethics section that licensing was handled appropriately. No additional content was introduced, and we kept the original license identifiers for every file.
> >
> > Regarding release, yes, **we intend to release the the VectorGym datasets used in our experiments.** The underlying data allow redistribution, and our processed dataset remains compatible with the original licenses.
> >
> > We have added this information in the manuscript (Appendix F)

---

### Author Response · Authors · 2025-11-27
**General Response to all Reviewers**

**We thank all reviewers for the detailed reviews and constructive feedback.** In response, we have made several focused changes to clarify contributions and improve technical rigor:

1. **Clarified positioning with respect to SVGenius, VGBench, SVGEditBench and others**, fully rewrote the related work section, and corrected all faulty citations and Table 1 entries, including explicit flags for human annotations and supervision types.

2. **Precisely defined “complex human annotations,”** detailed the **editing and sketch collection pipeline** (expert vendors, tools, and criteria), and **added concrete examples in the main text and appendix** to illustrate the level of difficulty and semantic reasoning involved.

3. **Expanded the benchmark description in Section 3,** including **dataset construction details, filtering thresholds, and the full VLM as judge setup**; added a dedicated subsection on **data licensing**, explaining that all SVGs come from The Stack under permissive licenses and that VectorGym will be released accordingly.

4. **Removed the best-of-n pipeline and reran all evaluations with single-shot decoding** to avoid circular scoring.

5. **Redesigned the VLM-judge prompt into a clearer 0 to 5 scale** and fully **reran the human evaluation and correlation study** with *more samples, more annotators, and more judges*.

6. **Added new baselines, including Gemini 3 Pro** and **strong open source judges (Qwen2.5VL, Qwen3VL 32B/235B, GLM4.5V)**, and **recomputed all tables** with these additions.

7. **Added new training experiments using GRPO on the VectorGym training split with a Qwen3VL 8B model**, showing that *our curated rewards produce a strong open source baseline that competes closely with closed models*.

We believe these revisions address the main concerns on novelty, methodology, evaluation, and licensing, and they substantially strengthen the clarity and reliability of the VectorGym benchmark.

**We are happy to discuss and clarify any remaining questions the reviewers have.**

Thanks again for your time.

*VectorGym authors*

---

### Meta-Review · Area_Chair_1QCz · 2026-01-14

**Summary:**

This paper introduces VectorGym, a human-annotated multi-task benchmark for evaluating Vision-Language Models (VLMs) on Scalable Vector Graphics (SVG) understanding, generation, and manipulation. The benchmark covers four tasks—Sketch2SVG, instruction-guided SVG editing, Text2SVG generation, and SVG captioning, supported by a curated dataset of approximately 7,000 real-world SVGs with accompanying sketches, captions, and annotations.

**Reviewer Concerns:**

Overall, the reviewers raised substantial concerns regarding the aspects of novelty, clarity, rigor, and credibility. Multiple reviewers also pointed out serious bibliographic and citation errors, including misattributed or nonexistent works, which undermine confidence in the literature review and claims of novelty. Methodologically, the dataset construction, task definitions, and editing procedures are considered to be insufficiently described, alongside unresolved legal/licensing questions about SVG data usage. Other concerns were raised about dataset contamination, questionable evaluation setups and the lack of analysis on fine-tuning or generalization effects.

**Reviewer Scores:**

As also summarized in the authors’ rebuttal, multiple reviewers raised concerns regarding the paper’s contributions and technical rigor. Overall, reviewers consistently rated the level of contribution as fair, and the aggregate evaluation tends to lean toward a negative recommendation.

---

### Decision · Program_Chairs · 2026-01-26

Reject